# Loss of Elp1 disrupts trigeminal ganglion neurodevelopment in a model of familial dysautonomia

Carrie E Leonard[1], Jolie Quiros[1], Frances Lefcort[2], Lisa A Taneyhill[1]*

[1]Department of Avian and Animal Sciences, University of Maryland, College Park, College Park, United States; [2]Department of Microbiology and Cell Biology, Montana State University, Bozeman, United States

**Abstract** Familial dysautonomia (FD) is a sensory and autonomic neuropathy caused by mutations in elongator complex protein 1 (*ELP1*). FD patients have small trigeminal nerves and impaired facial pain and temperature perception. These signals are relayed by nociceptive neurons in the trigeminal ganglion, a structure that is composed of both neural crest- and placode-derived cells. Mice lacking *Elp1* in neural crest derivatives ('*Elp1* CKO') are born with small trigeminal ganglia, suggesting Elp1 is important for trigeminal ganglion development, yet the function of Elp1 in this context is unknown. We demonstrate that Elp1, expressed in both neural crest- and placode-derived neurons, is not required for initial trigeminal ganglion formation. However, *Elp1* CKO trigeminal neurons exhibit abnormal axon outgrowth and deficient target innervation. Developing nociceptors expressing the receptor TrkA undergo early apoptosis in *Elp1* CKO, while TrkB- and TrkC-expressing neurons are spared, indicating Elp1 supports the target innervation and survival of trigeminal nociceptors. Furthermore, we demonstrate that specific TrkA deficits in the *Elp1* CKO trigeminal ganglion reflect the neural crest lineage of most TrkA neurons versus the placodal lineage of most TrkB and TrkC neurons. Altogether, these findings explain defects in cranial gangliogenesis that may lead to loss of facial pain and temperature sensation in FD.

## Editor's evaluation

This study uses a combination of conditional knockout mouse embryos with targeted deletion of Elp1 in neural crest cells and neuron-specific antibodies to identify the onset of neural defects associated with the trigeminal ganglion. This manuscript will be of interest to developmental biologists studying neurodevelopment disorders and provides important insights into the mechanisms underlying Familial Dysautonomia in the cranial sensory ganglia.

*For correspondence:
ltaney@umd.edu

## Introduction

Hereditary sensory and autonomic neuropathies (HSANs) are a group of phenotypically similar, yet distinct, peripheral nervous system disorders that stem from unique mutations (*Schwartzlow and Kazamel, 2019*). The most prevalent form is HSAN type III, or familial dysautonomia (FD), which is almost exclusively caused by a mutation in intron 20 of the elongator complex protein 1 (*ELP1*, formerly *IKBKAP*) gene (*Anderson et al., 2001*; *Slaugenhaupt et al., 2001*). This mutation causes mis-splicing and skipping of exon 20 in a tissue-specific manner, consequently reducing ELP1 protein in neurons (*Slaugenhaupt et al., 2001*; *Hims et al., 2007*). Elp1 is a requisite scaffolding protein of the six-subunit elongator complex that regulates translation by modifying particular tRNAs (*Huang et al., 2005*; *Huang et al., 2008*; *Karlsborn et al., 2014*; *Xu et al., 2015*; *Esberg et al., 2006*).

Deletion of elongator subunits has been shown to alter tRNA modifications and protein expression in several systems (*Cameron et al., 2021*; *Chen et al., 2009b*; *Esberg et al., 2006*; *Goffena et al., 2018*; *Karlsborn et al., 2014*; *Kojic et al., 2021*). Consequently, loss or reduction of Elp1 protein leads directly or indirectly to the clinical phenotypes seen in FD, which include impaired pain and temperature sensation, feeding and swallowing difficulties, blood pressure instability, tachycardia, optic neuropathy, and gastrointestinal dysfunction, among other symptoms. The disease is fatal, with the majority of patients dying by age 50 (*Gold-von Simson and Axelrod, 2006*). Autopsy reports, patient studies, and animal models demonstrate FD phenotypes are caused by abnormal neurode-velopment, in addition to neurodegeneration across the lifespan of the affected individual (*Pearson et al., 1978*; *Hunnicutt et al., 2012*; *George et al., 2013*; *Jackson et al., 2014*; *Li et al., 2020*; *Won et al., 2019*; *Lefcort et al., 2017*; *Dietrich and Dragatsis, 2016*).

FD severely impacts the peripheral nervous system, composed of sensory and autonomic neurons that are mostly derived from neural crest cells and some cranial sensory neurons that arise from ectodermal placodes (*Méndez-Maldonado et al., 2020*; *Steventon et al., 2014*; *Kameneva and Adameyko, 2019*). Since mice null for *Elp1* are embryonic lethal (*Chen et al., 2009a*; *Dietrich et al., 2011*), FD has been modeled in mice using conditional knockout of *Elp1* in neural crest cells (*Elp1*$^{fl/fl}$; *Wnt1-Cre*$^{+}$, abbreviated '*Elp1* CKO'), which recapitulates several aspects of the human disease, including significant loss of sensory and autonomic neurons (*George et al., 2013*; *Jackson et al., 2014*). To date, these studies have focused on mechanisms of Elp1 in sensory and sympathetic neurons in the trunk. However, FD patients also experience cranial sensory deficits such as impaired sensation of facial pain and temperature, neurogenic dysphagia, absent corneal reflexes, and reduced basal lacrimation (*Mendoza-Santiesteban et al., 2017*; *Palma et al., 2014*; *Barlow, 2009*; *Geltzer et al., 1964*; *Gutiérrez et al., 2015*; *Palma et al., 2018*). These functions rely on input from the trigeminal nerve, the largest of the cranial nerves, which contains axons of sensory neurons that reside in the trigeminal ganglion. A quantitative MRI study revealed FD patients have smaller trigeminal nerves compared to healthy, age-matched individuals, but there are no indications of progressive trigeminal nerve degeneration (*Won et al., 2019*). Moreover, trigeminal ganglia of *Elp1* CKO mice are reduced in size at birth compared to Controls (*Jackson et al., 2014*). Together, these findings suggest Elp1 may play an important role in the development of trigeminal sensory neurons, a critical aspect of FD that has yet to be explored.

While peripheral neurons in the trunk are exclusively neural crest-derived, cranial sensory neurons arise from two progenitor populations, neural crest cells and ectodermal placodes, which typically contribute neurons to spatially distinct regions of the cranial ganglia (*York et al., 2020*; *Moody and LaMantia, 2015*; *Park and Saint-Jeannet, 2010*; *Steventon et al., 2014*). The trigeminal ganglion is unique in that it contains intermixed neural crest- and placode-derived neurons that rely on one another for proper migration, coalescence, and function during development (*Blentic et al., 2011*; *D'Amico-Martel, 1982*; *D'Amico-Martel and Noden, 1983*; *Freter et al., 2013*; *Hamburger, 1961*; *Saint-Jeannet and Moody, 2014*; *Shiau et al., 2008*; *Steventon et al., 2014*). Shortly after differ-entiation, trigeminal ganglion neuron subtypes are discernable by mutually exclusive expression of tropomyosin receptor kinase (Trk) receptors, TrkA, TrkB, or TrkC, which are required for target inner-vation and long-term survival (*Huang et al., 1999a*; *Wilkinson et al., 1996*; *Scott-Solomon and Kuruvilla, 2018*; *Davies, 1997*; *Reichardt, 2006*; *Huang et al., 1999b*). Importantly, Trk expression generally correlates with the ultimate sensory modality encoded by a particular neuron; for example, small-diameter TrkA neurons are typically associated with pain and temperature perception, while large-diameter TrkB and TrkC neurons are usually mechanoreceptors that sense touch, pressure, and vibrations (*Mu et al., 1993*; *Genç et al., 2005*; *d'Amico-Martel and Noden, 1980*; *D'Amico-Martel and Noden, 1983*; *Davies and Lumsden, 1984*).

In dorsal root, epibranchial (sensory), and sympathetic (autonomic) ganglia, Elp1 is required for the generation and/or survival of TrkA and TrkB neurons, while TrkC neurons are spared during develop-ment (*George et al., 2013*; *Jackson et al., 2014*; *Tolman et al., 2022*). In these contexts, neuronal loss has been attributed to decreased neurogenesis resulting from early differentiation and apoptosis of progenitors (*George et al., 2013*; *Goffena et al., 2018*) and to severe target innervation defects and neuronal apoptosis due to insufficient neurotrophic support (*Jackson et al., 2014*; *Li et al., 2020*; *Tolman et al., 2022*). Notably, other less common forms of HSANs arise from germline mutations in the genes encoding either TrkA or its high-affinity ligand, nerve growth factor (NGF; *Schwartzlow*

and Kazamel, 2019). A growing body of evidence from animal models suggests Elp1 regulates TrkA signaling as well (*Abashidze et al., 2014*; *Lefler et al., 2015*; *Naftelberg et al., 2016*; *Li et al., 2020*), although the relationship between Elp1 and TrkA has not been investigated in the cranial ganglia. While there are clear trigeminal sensory deficits in FD, the function of Elp1 in the trigeminal ganglion and its nerves remains unexamined. Moreover, it is unknown whether certain neuronal subtypes within the trigeminal ganglion are more vulnerable to Elp1 loss than others, or whether trigeminal sensory phenotypes in FD patients arise from defects in neural crest-derived neurons, placodal neurons, or both.

Here, we describe the first comprehensive analysis of neurodevelopmental changes in the trigeminal ganglion using an established *Elp1* CKO mouse model of FD (*George et al., 2013*). We observe that Elp1 protein is enriched in the cytoplasm of differentiated neural crest- and placode-derived neurons in the trigeminal ganglion. In *Elp1* CKO mice, initial formation of the trigeminal ganglion appears unaltered, but trigeminal nerve growth is severely diminished as development proceeds. Innervation deficits correlate with decreased TrkA levels in the peripheral and central projections of trigeminal ganglion neurons and a significant reduction in the number of TrkA neurons due to early apoptosis. Finally, we demonstrate that the majority of TrkA neurons in the trigeminal ganglion are neural crest-derived, whereas the majority of TrkB and TrkC neurons are placode-derived, explaining the specific TrkA defects in the neural crest-targeted *Elp1* CKO. Collectively, these findings indicate Elp1 is required for proper target innervation and survival of neural crest-derived TrkA neurons in the trigeminal ganglion. Moreover, the use of a neural crest-specific knockout sheds light on the lineage and dynamics of neuronal subpopulations in the trigeminal ganglion. Importantly, our findings reveal novel neurodevelopmental defects in cranial gangliogenesis that may ultimately contribute to facial sensory deficits experienced by patients with FD.

## Results

### Elp1 protein is enriched in the cytoplasm of trigeminal ganglion sensory neurons

While ubiquitous Elp1 expression in rodent embryos has been previously reported (*Mezey et al., 2003*; *George et al., 2013*), the spatiotemporal distribution of Elp1 in the craniofacial complex had not been evaluated. To initially address this, we examined the *Elp1^LacZ* (previously '*Ikbkap:LacZ*') reporter mouse that expresses β-galactosidase from the *Elp1* locus (*George et al., 2013*). Whole-mount preparations uncovered widespread β-galactosidase staining in the head at embryonic day 10.5 (E10.5), with prominent expression in the neural tube, regions of the developing cranial ganglia, the pharyngeal arches, and the facial prominences (*Figure 1A*). Horizontal sections through the head at E10.5 revealed diffuse β-galactosidase expression in the trigeminal ganglion, which, at this stage, is a newly condensed structure containing differentiated trigeminal placode-derived neurons and undifferentiated cranial neural crest cells (*Figure 1B and C*, *Karpinski et al., 2016*). β-galactosidase was also robustly expressed in the neural tube and neuroretina and scattered throughout the cranial mesenchyme (*Figure 1B and C*). Together, these results indicate that the *Elp1* gene is expressed in cranial neural tissues, including the newly formed trigeminal ganglion.

Detection of Elp1 via immunohistochemistry revealed a similar expression pattern to that found in the *Elp1^LacZ* reporter; however, Elp1 protein was more discretely expressed compared to β-galactosidase. At E10.5, Elp1 protein was abundant in the cytoplasm of differentiated neurons, which were identified by expression of the transcription factor Islet1 (*Figure 1D–G* and *Figure 1—figure supplement 1A-C*). Since the majority of neurons in the trigeminal ganglion are placode-derived at this stage, while surrounding neural crest cells have yet to differentiate (*Karpinski et al., 2016*), it appears that Elp1 is initially expressed in trigeminal placode-derived neurons. In contrast, condensed neural crest cells within the trigeminal ganglion, identified by expression of the surface receptor Neuropilin 2 or the transcription factor Pax3, and devoid of Islet1, expressed little to no Elp1 protein (*Figure 1E* and *Figure 1—figure supplement 1A-D*).

At E11.5, when neural crest-derived neurons are also present in the trigeminal ganglion, Elp1 protein was observed in all differentiated neurons expressing β-tubulin III (Tubb3) or Islet1, irrespective of their developmental origin, with the highest signal detected in axons (*Figure 1H–M* and *Figure 1—figure supplement 1E-H*). By E12.5, enrichment of Elp1 protein in the neuronal cytoplasm, especially

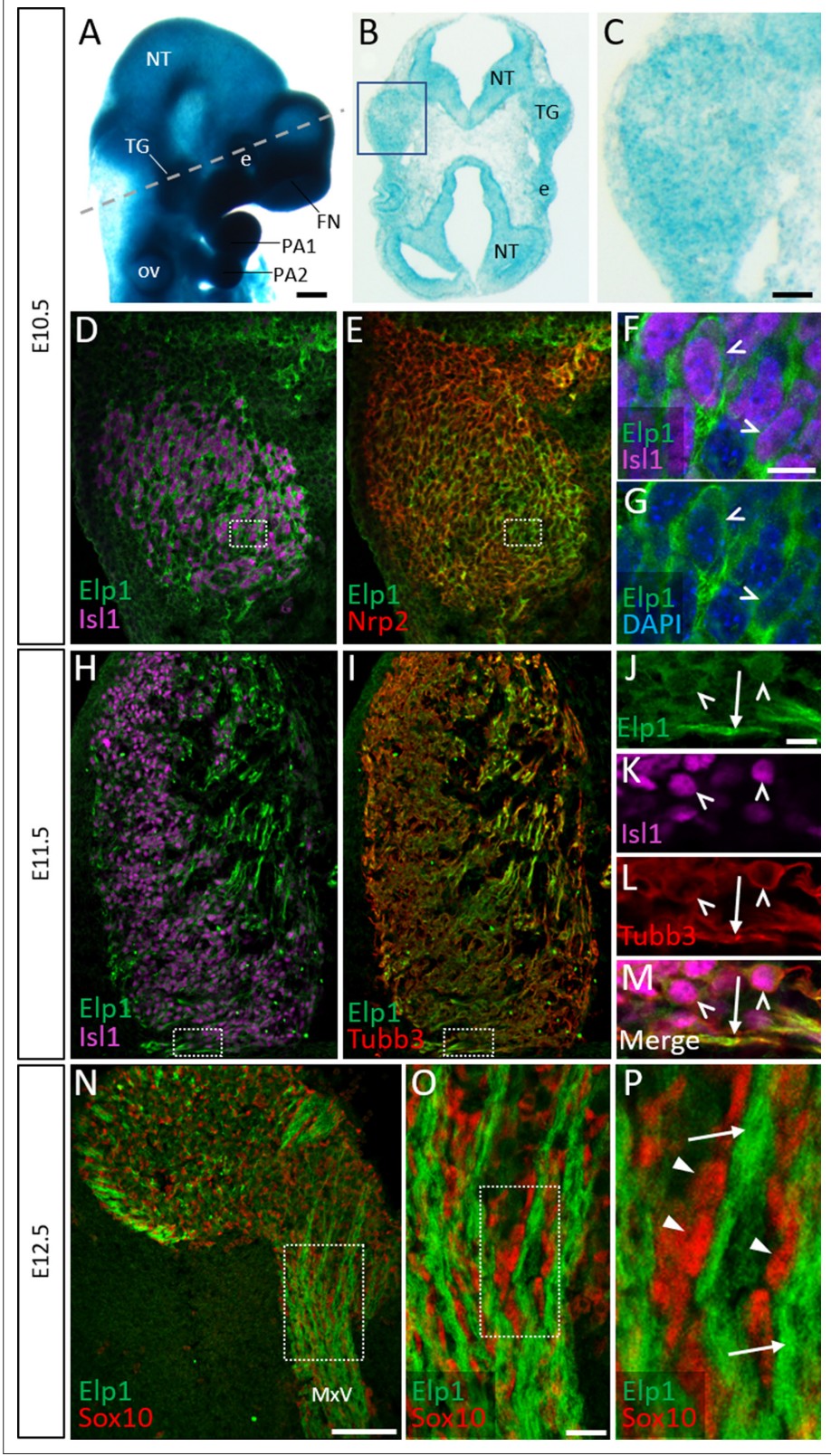

**Figure 1.** Elp1 protein is enriched in the cytoplasm of developing trigeminal ganglion neurons. (**A**) Lateral view of embryonic day 10.5 (E10.5) *Elp1^LacZ* reporter mouse stained for β-galactosidase. Dashed line indicates the plane of section for the same embryo shown in B. (**B and C**) Representative horizontal section through embryo in A to reveal *Elp1* gene expression. Boxed region in B is magnified and shown in C. (**D–P**) Representative horizontal

*Figure 1 continued on next page*

*Figure 1 continued*

sections taken from Control E10.5 (D–G), E11.5 (H–M), or E12.5 (N–P) mouse embryos followed by fluorescent immunohistochemistry for Elp1 (D–P, green), Islet1 (D, F, H, K and M, 'Isl1', purple), Neuropilin 2 (E, 'Nrp2', red), β-tubulin III (I, L, M, 'Tubb3', red), and Sox10 (N–P, red). Boxed region in D and E is magnified in F–G and shows Elp1 (green), Isl1 (purple), and DAPI-stained nuclei (blue). (J–M) Higher magnification of box in H and I. (O) Higher magnification of box in N. (P) Higher magnification of box in O. Carets indicate Isl1-positive neuronal nuclei (F, G, J–M) and/or Tubb3-positive neuronal cell bodies (J–M). Arrows identify axons (J, L, M, and P), while arrowheads point to Sox10-positive glial progenitors (P). Abbreviations: e: eye; FN: frontonasal prominence; NT: neural tube; ov: otic vesicle; PA: pharyngeal arch; TG: trigeminal ganglion. Scale bars: 400 µm (A), also applies to B; 50 µm (C), applies to D, E, H, and I; 100 µm (N); 10 µm (F), applies to G; 10 µm (J), applies to K, L, and M; 20 µm (O), applies to P as 5 µm.

The online version of this article includes the following figure supplement(s) for figure 1:

**Figure supplement 1.** Elp1 protein is not enriched in Pax3-positive neural crest cells or glial progenitors in the developing trigeminal ganglion.

---

axons, is still apparent, while Elp1 is absent or expressed at relatively low levels in neural crest-derived Sox10-positive glia (*Figure 1N–P*). Collectively, these results suggest important functions for Elp1 in developing sensory neurons of the trigeminal ganglion.

## Neural crest-specific deletion of *Elp1* causes progressive morphological abnormalities in the trigeminal ganglion and nerves

To gain insight into the etiology of FD phenotypes associated with trigeminal ganglion dysfunction, an established mouse model was used, in which *Elp1* is deleted from neural crest cells and their derivatives via Wnt1-Cre-mediated recombination. *Elp1* conditional knockouts (*Wnt1-Cre⁺; Elp1^{flox/flox}*, 'Elp1 CKO') were compared to littermate Controls (*Wnt1-Cre⁻; Elp1^{flox/+}*), with at least two litters analyzed per experiment (*George et al., 2013*). To evaluate neurodevelopmental dynamics in Control and *Elp1* CKO, we visualized intact trigeminal ganglia and developing nerves at intervals ranging from early (E10.5) to later (E12.5–13) neurogenic stages, using whole-mount Tubb3 immunohistochemistry to label all neurons. The *Elp1* CKO trigeminal ganglion and nerve phenotypes relative to Controls are summarized in *Table 1*.

**Table 1.** Comparison of *Elp1* CKO cranial ganglia and nerve phenotypes relative to Control over developmental time.

Summary of observations of the trigeminal ganglion, ophthalmic nerve, maxillary nerve, mandibular nerve, central nerve root, and geniculate ganglion in *Elp1* CKO between embryonic day 10.5 (E10.5) and E13, as compared to Control littermates. *, mandibular nerve was difficult to visualize at these stages, so no observations were recorded.

| | E10.5 | E11.5 | E12.5–13 |
|---|---|---|---|
| Trigeminal ganglion | No difference in size; Isl1/Six1+ placodal neurons and Sox10+ neural crest cells present in similar numbers | No difference in size | Slightly smaller size (not statistically significant); increased TUNEL staining; fewer TrkA neurons; decreased TrkA immunoreactivity |
| Ophthalmic nerve | Present; no difference in length | No difference | Wandering axons; decreased branching complexity; nasal nerve absent; less TrkA immunoreactivity |
| Maxillary nerve | Present | Disorganized and defasciculated axons | Reduced whisker pad innervation by infraorbital nerve; less TrkA immunoreactivity; fewer TrkA + nerve endings in target region |
| Mandibular nerve | Present; no difference in length | Disorganized and defasciculated axons | * |
| Central root | No difference | No difference | Significantly smaller; reduced TrkA immunoreactivity |
| Geniculate ganglion | No difference in size | No difference in length of chorda tympani nerve | No increased TUNEL staining |

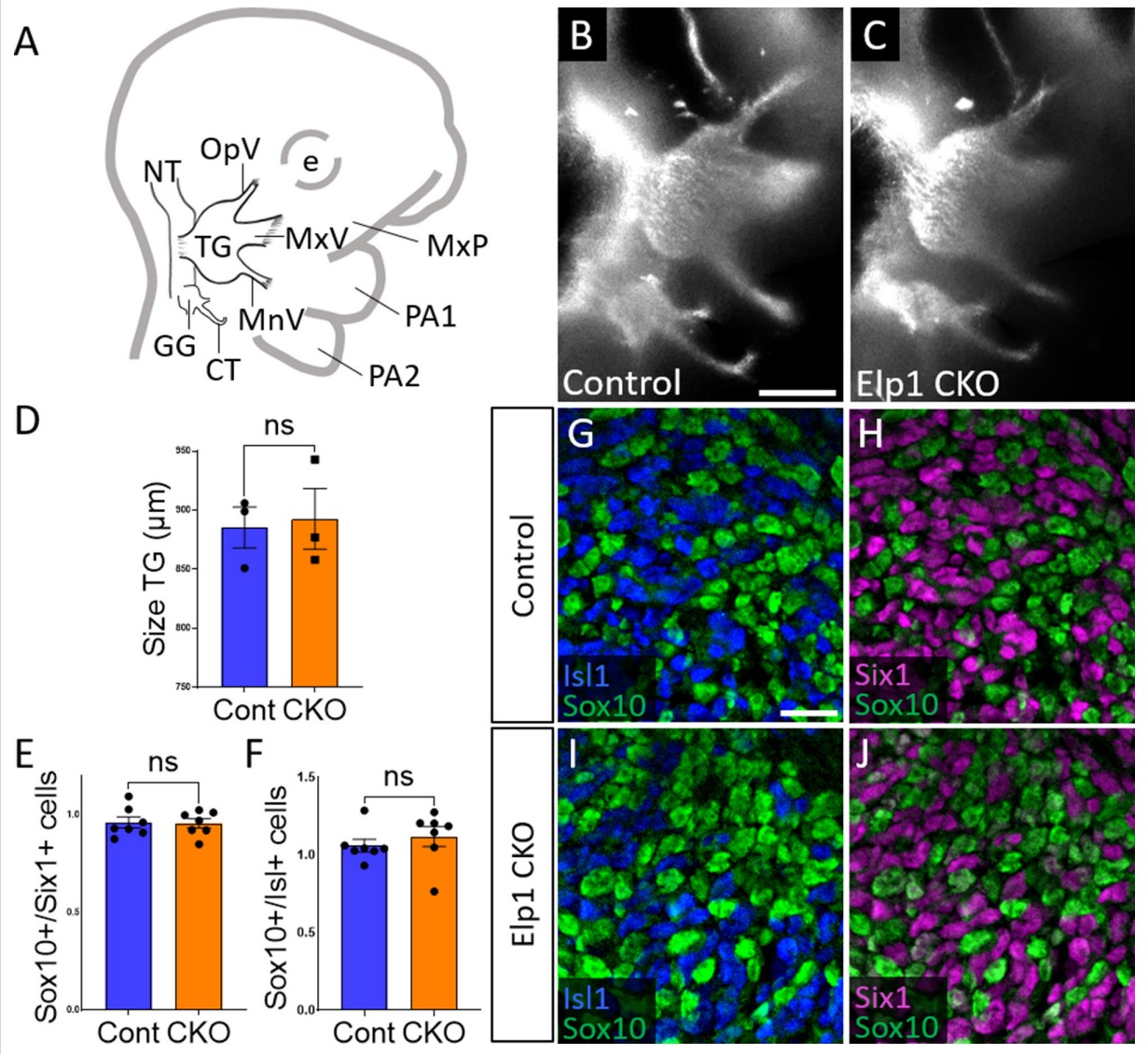

**Figure 2.** Initial trigeminal ganglion formation appears normal in *Elp1* CKO at embryonic day 10.5 (E10.5). (**A**) Schematic depicting relevant cranial anatomy in E10.5 mouse: CT: chorda tympani nerve; e: eye; GG: geniculate ganglion; MnV: mandibular nerve; MxP: maxillary process; MxV: maxillary nerve; NT: neural tube; OpV: ophthalmic nerve; PA: pharyngeal arch; and TG: trigeminal ganglion. (**B and C**) Lateral view of trigeminal and geniculate ganglia in Control and *Elp1* CKO (*Wnt1-Cre⁺;Elp1flox/flox*) littermates after Tubb3 whole-mount immunohistochemistry (white) to label neurons. (**D**) Quantification of the size of the TG in Control (blue, 885.3 µm, n=3) and *Elp1* CKO (orange, 892.7 µm, n=3, p=0.8247, unpaired t-test with Holm-Sidak correction for multiple comparisons). (**E**) Quantification showing the ratio of Sox10-positive cells to Six1-positive cells in Control (blue, 0.96, n=3) and *Elp1* CKO (orange, 0.94, n=3, p=0.8232, nested unpaired t-test adjusted for multiple comparisons). (**F**) Quantification showing the ratio of Sox10-positive cells to Isl1-positive cells in Control (blue, 1.04, n=3) and *Elp1* CKO (orange, 1.13, n=3, p=0.5033). Values for histograms represent mean ± SEM. (**G–J**) Fluorescent immunohistochemistry on representative horizontal sections through the TG from Control (G and H) or *Elp1* CKO (I and J) littermates shows placodal neurons labeled by Isl1 (G and I, blue) or Six1 (H and J, purple) and neural crest cells labeled by Sox10 (G–J, green). Scale bars: 400 µm (B), also applies to C; 20 µm (G), applies to H–J. Refer to *Figure 2—source data 1* for quantitative summary data represented in graphs.

The online version of this article includes the following source data for figure 2:

**Source data 1.** Initial trigeminal ganglion (TG) formation appears normal in Elp1 CKO at embryonic day 10.5 (E10.5).

When compared to littermate Controls at E10.5, initial formation of the trigeminal ganglion appeared normal in *Elp1* CKO (*Figure 2B and C*). Gross anatomy was intact with no significant difference in the size of the trigeminal ganglion (*Figure 2D*). The ophthalmic, maxillary, and mandibular nerve branches were all present in both Control and *Elp1* CKO (*Figure 2B and C*), with no significant changes in the length of the ophthalmic nerve (618.7 µm Control, n=3, versus 569.0 µm *Elp1*

CKO, n=3, p=0.0520, unpaired t-test, Holm-Sidak adjustment for multiple comparisons) nor the mandibular nerve (896.3 µm Control, n=3, versus 905.0 µm *Elp1* CKO, n=3, p=0.832, unpaired t-test, Holm-Sidak adjustment for multiple comparisons). Immunohistochemistry on E10.5 tissue sections revealed placodal neurons (Islet1-positive and Six1-positive nuclei) and undifferentiated neural crest cells (Sox10-positive nuclei) present in similar ratios throughout the forming trigeminal ganglion (*Figure 2E–J*). There was also no change in the size or length of the exclusively placode-derived geniculate ganglion (387 µm Control, n=3, versus 361 µm *Elp1* CKO, n=3, p=0.4559, unpaired t-test, Holm-Sidak adjustment for multiple comparisons) or chorda tympani nerve (692 µm Control, n=3, versus 686.3 µm *Elp1* CKO, n=3, p=0.7671, unpaired t-test, Holm-Sidak adjustment for multiple comparisons), whose neurons are not targeted by Wnt1-Cre for *Elp1* deletion (*Figure 2B and C*). Altogether, these data suggest Elp1 is not required in neural crest cells for them to coalesce with placodal neurons as the trigeminal ganglion is initially forming.

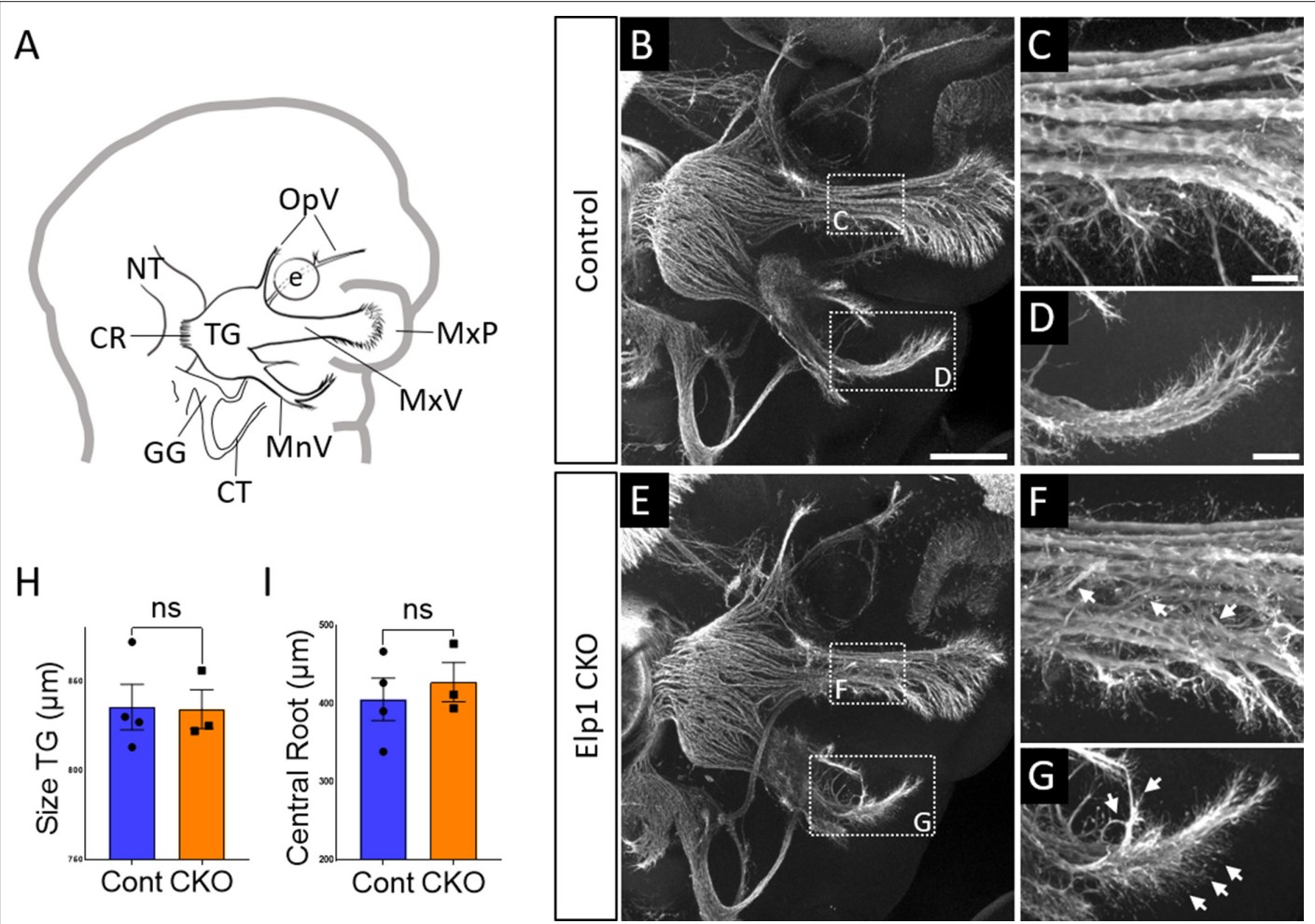

**Figure 3.** Progressive trigeminal nerve abnormalities arise in *Elp1* CKO starting at embryonic day 11.5 (E11.5). (**A**) Schematic depicting relevant cranial anatomy in E11.5 mouse: CR: central root; CT: chorda tympani nerve; e: eye; GG: geniculate ganglion; MnV: mandibular nerve; MxP: maxillary process; MxV: maxillary nerve; NT: neural tube; OpV: ophthalmic nerve; and TG: trigeminal ganglion. (**B–G**) Representative maximum intensity projections of confocal Z-stacks through Control (B–D) or *Elp1* CKO (E–G) littermates, which were processed for whole-mount immunohistochemistry to detect Tubb3 (white), followed by tissue clearing. (C, D, F, and G) Higher magnification of boxes in B and E. Arrows indicate disorganized axons (F and G) in *Elp1* CKO. (**H**) Quantification of the size of the TG in Control (blue, 835.5 µm, n=4) and *Elp1* CKO (orange, 834.3 µm, n=3, p=0.9497, unpaired t-test with Holm-Sidak adjustment for multiple comparisons). (**I**) Quantification of the central root diameter in Control (blue, 405.9 µm, n=4) and *Elp1* CKO (orange, 427.7 µm, n=3, p=0.9497, unpaired t-test adjusted for multiple comparisons). Values for histograms represent mean ± SEM. Scale bar: 200 µm (B), applies to E; also applies to C, D, F, and G as 50 µm. Refer to *Figure 3—source data 1* for quantitative summary data represented in graphs.

The online version of this article includes the following source data for figure 3:

**Source data 1.** Progressive trigeminal nerve abnormalities arise in Elp1 CKO starting at embryonic day 11.5 (E11.5).

By E11.5, Control and *Elp1* CKO displayed robust outgrowth of axons from the trigeminal ganglion, which contains both neural crest- and placode-derived neurons at this stage (*Figure 3B and E*). Although the ophthalmic, maxillary, and mandibular nerves were present and distinct in Control and *Elp1* CKO, nerves in the latter appeared less organized. In the maxillary nerve, *Elp1* CKO axons often strayed from established bundles into the surrounding mesenchyme or crossed to adjacent fascicles (*Figure 3C and F*). Axons of the mandibular nerve in *Elp1* CKO also traveled away from the established nerve without direction, compared to the compact mandibular nerve observed in Controls (*Figure 3D and G*). The *Elp1* CKO phenotype is highly penetrant, and disrupted axon trajectories were noted to varying degrees in all *Elp1* CKO embryos examined. In contrast, the placode-derived chorda tympani nerve exhibited normal axon trajectories in both Control and *Elp1* CKO embryos at E11.5 (*Figure 3B and E*). At this stage, there was no significant difference in the size of the trigeminal ganglion or the width of the central nerve root, which contains axons projecting from the trigeminal ganglion into the hindbrain, in Control versus *Elp1* CKO (*Figure 3H and I*). Additionally, there were similar ratios of Sox10-positive cells (Control = 47.1%, n=3, versus *Elp1* CKO = 50.3%, n=3, of all DAPI-stained nuclei, p=0.6444, nested two-tailed t-test) and Six1-positive cells (Control = 69.7%, n=3, versus *Elp1* CKO = 65.9%, n=3, of all DAPI-stained nuclei, p=0.4897, nested t-test) in Control and *Elp1* CKO trigeminal ganglia. Therefore, while the E11.5 *Elp1* CKO trigeminal ganglion appeared generally intact, altered axon trajectories of trigeminal sensory neurons became evident at this stage.

By E12.5, trigeminal ganglion and nerve morphology were drastically altered in *Elp1* CKO, with nuanced changes in specific nerve branches (*Figure 4*). The size of the trigeminal ganglion proper trended smaller but was not statistically significant (*Figure 4B, E and H*). Interestingly, the diameter of the nerve root was significantly smaller in *Elp1* CKO, indicating the loss of some centrally projecting axons between E11.5 and E12.5 (*Figure 4B, E and I*). The developing frontal nerve (ophthalmic division) was present in both Control and *Elp1* CKO; however, the branching complexity of the nerve was decreased in *Elp1* CKO, as quantified by a modified Sholl analysis (*Figure 4C, F, J, and K*, *Sholl, 1953*). Stray frontal nerve axons were also observed proximal to the ganglion (*Figure 4F*). While the ophthalmic nerve extended rostrally in both Control and *Elp1* CKO, the medial and lateral nasal nerves that descend from this branch toward the nose were absent in *Elp1* CKO (*Figure 4D and G*). The maxillary division was also impacted, such that the terminal extent of the infraorbital nerve, which innervates the whisker pad, was smaller in *Elp1* CKO compared to Control (*Figure 4D, G and L*). Together, these results provide a dynamic summary of the requirements for Elp1 in axonal pathfinding and target innervation in trigeminal ganglion sensory neurons. Importantly, our data suggest that nerve defects caused by loss of Elp1 can differ, even within the same ganglion, based on neuronal identity and/or specific target region.

## TrkA-expressing neurons are specifically vulnerable in *Elp1* CKO trigeminal ganglia

Given that trigeminal nerve growth is disrupted in *Elp1* CKO, we examined the different Trk-expressing neuronal populations in Control and *Elp1* CKO trigeminal ganglia. First, the distribution of TrkA expression relative to Tubb3-labeled neurons was evaluated via whole-mount immunohistochemistry at E13. In Control embryos, TrkA expression was strong throughout the trigeminal ganglion and along all three major nerve divisions, detectable into the most distal branches of the frontal and infraorbital nerves (*Figure 5A, B, E, and F*, and *Figure 5—figure supplement 1*). In *Elp1* CKO, TrkA immunoreactivity was drastically reduced in the central nerve root (*Figure 5—figure supplement 1B,D*) and in distal branches of the frontal nerve, the rostral ophthalmic nerve, and the infraorbital nerve (*Figure 5C, D, G, and H*), indicating effects on both central and peripheral trigeminal projections.

Examination of E12.5 tissue sections through the maxillary lobe revealed significantly reduced TrkA immunofluorescence in *Elp1* CKO trigeminal ganglia compared to Control, with no change in TrkB or TrkC immunofluorescence (*Figure 5I–L and Q* and *Figure 5—figure supplement 1E-L*). Sections through the upper lip, which is heavily innervated by the infraorbital nerve, demonstrated fewer TrkA-expressing nerve endings in *Elp1* CKO compared to Control, while TrkC-expressing nerve endings were maintained (*Figure 5M–P*). To determine whether this loss of TrkA expression and innervation could be due to fewer TrkA neurons present at E12.5, each neuronal subtype was quantified within the maxillary lobe of the trigeminal ganglion. While no differences were observed in the number of TrkB or TrkC neurons, TrkA neurons were reduced by 23% in *Elp1* CKO compared to Controls

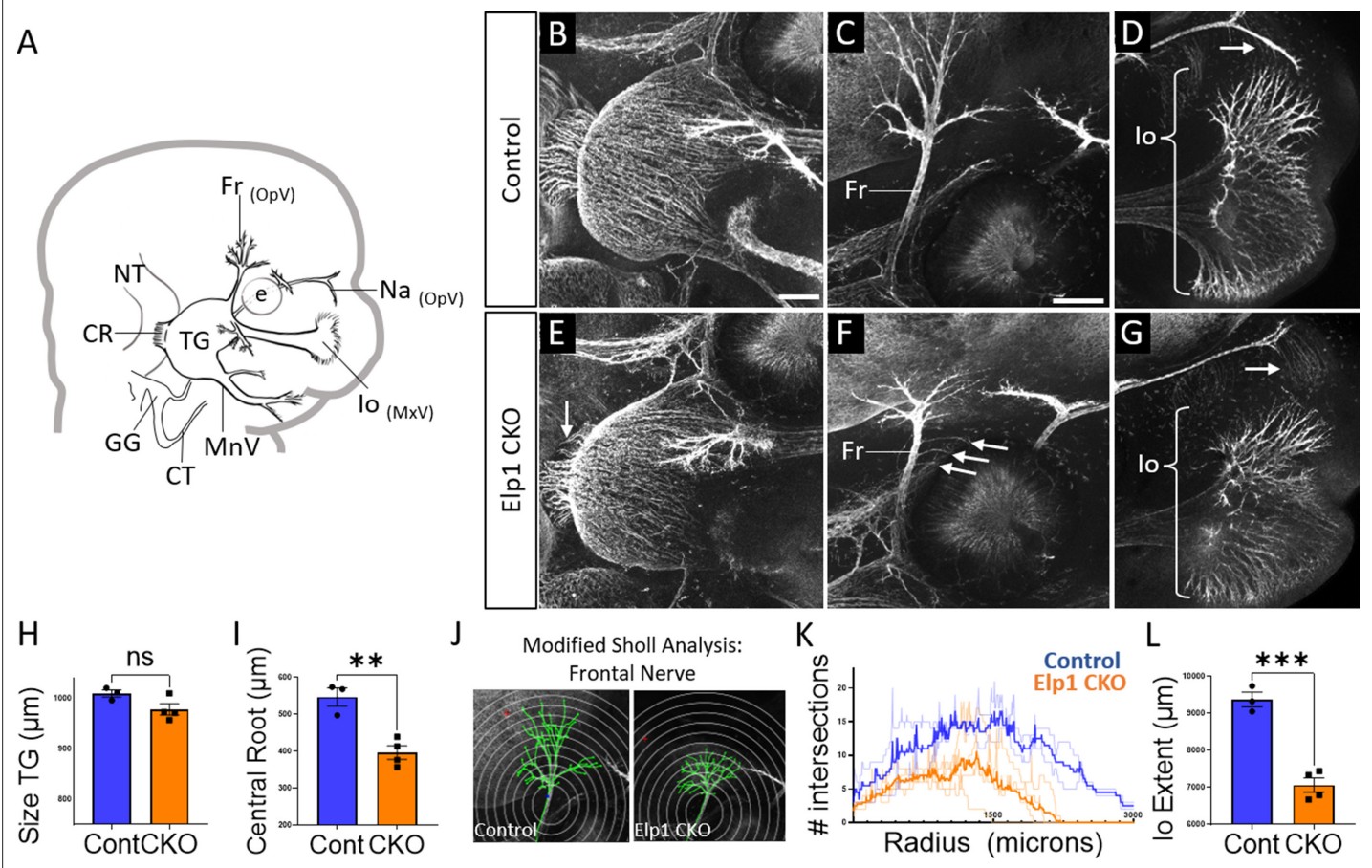

**Figure 4.** Trigeminal nerve branches are less complex or absent in *Elp1* CKO at embryonic day 12.5 (E12.5). (**A**) Schematic depicting relevant cranial anatomy in E12.5 mouse: CR: central root; CT: chorda tympani nerve; e: eye; Fr: frontal nerve; GG: geniculate ganglion; Io: infraorbital nerve; MnV: mandibular nerve; MxV: maxillary nerve; Na: nasal nerve; NT: neural tube; OpV: ophthalmic nerve; and TG: trigeminal ganglion. (**B–G**) Representative maximum intensity projections of confocal Z-stacks through Control (B–D) or *Elp1* CKO (E–G) littermates, which were processed for whole-mount immunohistochemistry to detect Tubb3 (white), followed by tissue clearing. (C, D, F, and G) Higher magnification images of the frontal nerve (C and F) or the infraorbital nerve at the developing whisker pad (D and G). Arrows indicate small central root (E), disorganized axons (F), and the absence of the nasal nerve (G) in *Elp1* CKO. (**H**) Quantification of the size of the TG in Control (blue, 1010 µm, n=3) and *Elp1* CKO (orange, 978.1 µm, n=4, p=0.0828, unpaired t-test with Holm-Sidak adjustment for multiple comparisons). (**I**) Quantification of the central root diameter in Control (blue, 546.9 µm, n=3) and *Elp1* CKO (orange, 396.8 µm, n=4, p=0.0828, unpaired t-test adjusted for multiple comparisons). (**J**) Diagram explaining modified Sholl analysis, with concentric circles of increasing radii overlayed on representative traces (green) of the frontal nerve in Control (left) or *Elp1* CKO (right). (**K**) Graph of modified Sholl analysis to quantify complexity of frontal nerve. Individual distributions are plotted in light blue (Control, n=2) and light orange (*Elp1* CKO, n=4), while group averages are plotted in dark blue (Control) and dark orange (*Elp1* CKO). (**L**) Quantification of the infraorbital nerve extent in Control (blue, 9374 µm, n=3) and *Elp1* CKO (orange, 7061 µm, n=4, p=0.0004, unpaired t-test adjusted for multiple comparisons). Values for histograms represent mean ± SEM. Scale bar: 200 µm (B), applies to E; 200 µm (C), applies to D, F, and G. Refer to *Figure 4—source data 1* for quantitative summary data represented in graphs.

The online version of this article includes the following source data for figure 4:

**Source data 1.** Trigeminal nerve branches are less complex or absent in Elp1 CKO at embryonic day 12.5 (E12.5).

(*Figure 5R*). Therefore, early target innervation defects in *Elp1* CKO trigeminal ganglion neurons are associated with a specific loss of TrkA-expressing neurons in the ganglion and nerves. These results are particularly intriguing since TrkA neurons typically function in nociception, and FD patients experience impaired sensation of facial pain and temperature.

Given that trigeminal ganglion neurons arise from both neural crest and placodal precursors, combined with our use of a neural crest-specific *Elp1* knockout, we assessed whether TrkA-specific deficits in *Elp1* CKO were correlated with *Elp1* deletion. Immunohistochemistry on E12.5 tissue sections revealed Elp1 protein expression throughout the Control trigeminal ganglion in TrkA, TrkB,

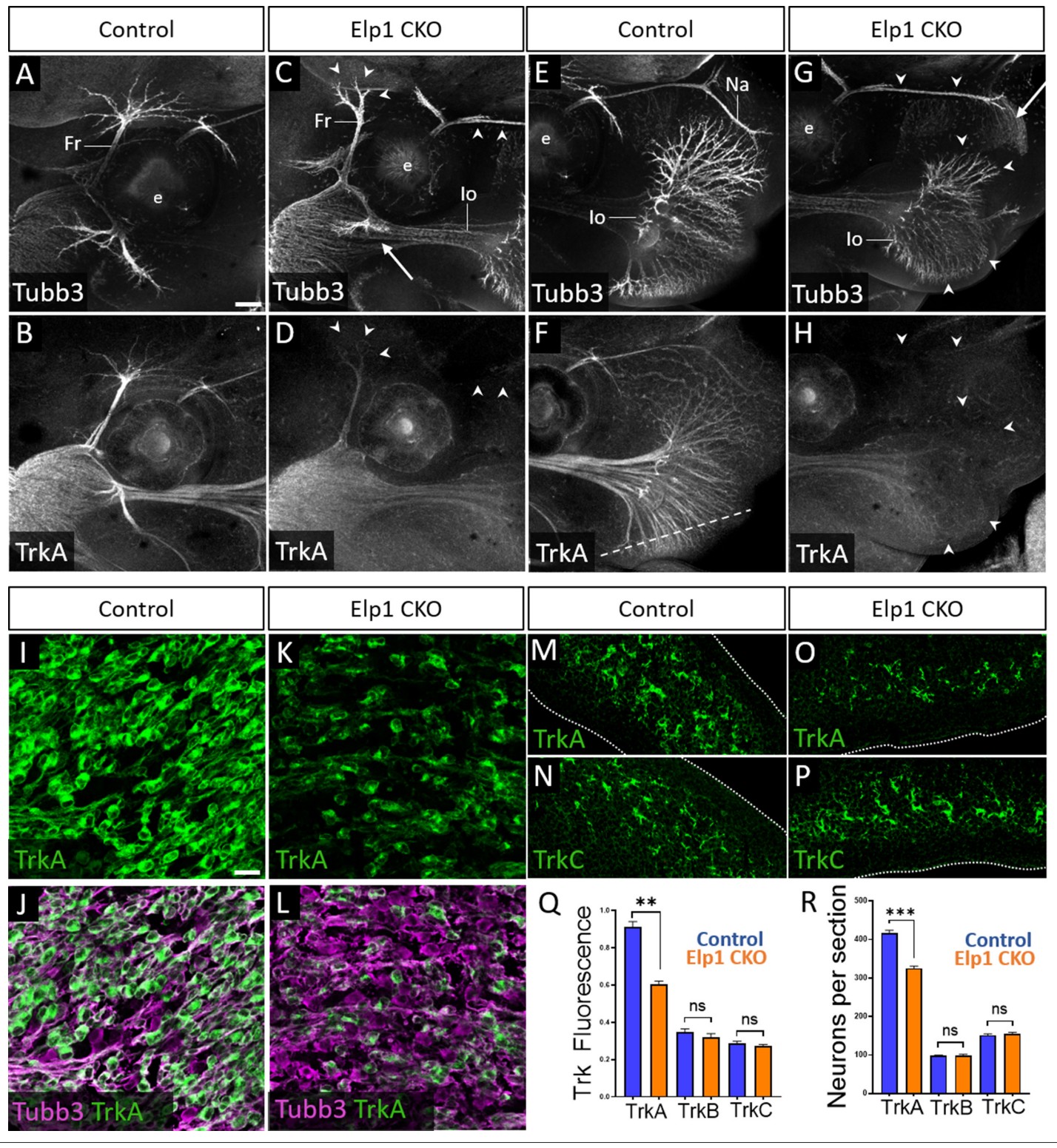

**Figure 5.** Loss of TrkA neurons with persistent innervation defects in *Elp1* CKO at embryonic day 12.5–13 (E12.5–13). (**A–H**) Representative maximum intensity projections of confocal Z-stacks through Control (A, B, E, and F) or *Elp1* CKO (C, D, G, and H) littermates, which were processed for whole-mount immunohistochemistry to detect Tubb3 (A, C, E, and G, white) and TrkA (B, D, F, and H, white), followed by tissue clearing. Arrows indicate regions where nerves are absent or severely diminished in *Elp1* CKO (C and G), while arrowheads point to areas with Tubb3-positive nerves but undetectable TrkA expression (C, D, G, and H). (**I–P**) Fluorescent immunohistochemistry on serial horizontal sections showing TrkA (I–M and O, green), Tubb3 (J and L, purple), or TrkC (N and P, green) in the trigeminal ganglion (I-L) or whisker pad (M–P) of Control (I, J, M, and N) or *Elp1* CKO (K, L,

*Figure 5 continued on next page*

*Figure 5 continued*

O, and P) littermates. Dashed line in F demonstrates the plane of section for M-P. (**Q**) Quantification of Trk fluorescent signal normalized to Tubb3 fluorescent signal within the trigeminal ganglia of Control (blue, TrkA = 0.897, TrkB = 0.3258, TrkC = 0.3047, n=3) and *Elp1* CKO (orange, TrkA = 0.6014, TrkB = 0.3551, TrkC = 0.2783, n=3, p=0.0037 TrkA, 0.4878 for TrkB, 0.7750 for TrkC, nested unpaired t-test with Holm-Sidak adjustment for multiple comparisons). (**R**) Quantification of Trk-expressing neurons in trigeminal ganglion sections of Control (blue, TrkA = 416.1, TrkB = 98.35, TrkC = 151.6, n=3) and *Elp1* CKO (orange, TrkA = 325.0, TrkB = 99.05, TrkC = 154.8, n=3, p=0.0150 for TrkA, 0.8966 for TrkB, 0.7675 for TrkC, nested unpaired t-test with Holm-Sidak adjustment for multiple comparisons). Values for histograms represent mean ± SEM. Scale bars: 200 μm (A), applies to (B–H); 20 μm (I), applies to (J–L) and applies to (M–P) as 25 μm. Refer to *Figure 5—source data 1* for quantitative summary data represented in graphs.

The online version of this article includes the following source data and figure supplement(s) for figure 5:

**Source data 1.** Loss of TrkA neurons with persistent innervation defects in Elp1 CKO at embryonic day 12.5–13 (E12.5–13).

**Figure supplement 1.** Loss of TrkA neurons is not accompanied by changes in TrkB and TrkC neuron subpopulations in *Elp1* CKO.

**Figure supplement 2.** Elp1 expression is retained in TrkB and TrkC neurons in *Elp1* CKO.

and TrkC neurons (*Figure 5—figure supplement 2A-C,G-I,M-O*). In contrast, most TrkA neurons in *Elp1* CKO were devoid of Elp1 (*Figure 5—figure supplement 2D-F*), while the majority of remaining Elp1-positive neurons expressed TrkB or TrkC (*Figure 5—figure supplement 2J-L,P-R*). Therefore, it seemed plausible that TrkA neurons are targeted in the *Elp1* CKO because TrkA neurons are the primary neural crest-derived neuronal population in the trigeminal ganglion.

## Neural crest and placode lineages are biased toward distinct Trk-expressing populations in the embryonic trigeminal ganglion

In light of the specific effects on TrkA neurons in *Elp1* CKO, which lacks *Elp1* in neural crest but not placode derivatives, we asked whether Trk-expressing subpopulations have distinct cellular precursors, which could explain the divergent effects on TrkA versus TrkB/C neurons. First, we examined the dynamics of normal neurogenesis and nerve growth in the trigeminal ganglion, with a focus on the maxillary division. Consistent with reports that TrkB and TrkC neurons are born first in the trigeminal ganglion followed by TrkA neurons (*Huang et al., 1999a*; *Huang et al., 1999b*), immunohistochemistry on trigeminal ganglion serial sections revealed that all three Trk subtypes are present at E11, but the majority of established infraorbital nerve endings are either TrkB- or TrkC-positive, with little contribution from TrkA-expressing axons at this stage (*Figure 6—figure supplement 1A-C*). By E11.5, some TrkA-positive axons have reached the whisker pad region, but the majority of infraorbital nerve endings are TrkA-negative and presumably TrkB- or TrkC-positive (*Figure 6—figure supplement 1D,E*). At this stage, TrkC neurons comprise the largest proportion of neurons, followed closely by TrkA, and TrkB representing the smallest proportion of trigeminal ganglion neurons (*Figure 6—figure supplement 1F-H*, *Huang et al., 1999a*; *Huang et al., 1999b*). In accordance with previous reports, the number of TrkB and TrkC neurons did not appear to increase between E11.5 and E12.5, but there was a robust increase in the number of TrkA neurons, such that TrkA neurons became the vast majority of trigeminal ganglion neurons by E12.5 (*Figure 6—figure supplement 1I-K*; *Huang et al., 1999b*). Given that neural crest deletion of *Elp1* targets TrkA neurons, and placode-derived neurons differentiate before neural crest-derived neurons, it follows that most TrkA neurons in the trigeminal ganglion are primarily neural crest-derived, while TrkB and TrkC neurons are placode-derived.

To validate these presumptions, we co-labeled for the transcription factor Six1, which is traditionally used as a marker of placode-derived neurons in the trigeminal ganglion (*Karpinski et al., 2016*; *Karpinski et al., 2022*; *Moody and LaMantia, 2015*; *Motahari et al., 2020*). At E10.5, we observed that 88.6% of TrkA neurons, 78.0% of TrkB neurons, and 91.8% of TrkC neurons expressed Six1 (*Figure 6A, D, G and J*). Surprisingly, at E11.5 and E12.5, the majority of Six1-expressing cells were TrkA neurons: 85.9% of TrkA neurons expressed Six1 at E11.5, while only 6.8% of TrkB neurons and 18.6% of TrkC neurons did (*Figure 6B, E, H, and J*). Similar results were noted at E12.5, with 50.6% of TrkA neurons expressing Six1, while only 1.8% of TrkB neurons and 6.8% of TrkC neurons expressed Six1 (*Figure 6C, F, I, and J*). When considering the timeline of TrkA versus TrkB/TrkC neurogenesis (*Figure 6—figure supplement 1*), these data suggest Six1 is a marker of recently differentiated neurons in the trigeminal ganglion, rather than an exclusive marker of the placodal lineage.

To confirm that Six1 labels newly differentiated neurons of both placode and neural crest origin, we examined Six1 expression in trigeminal ganglia of a neural crest lineage reporter mouse (*Wnt1-Cre;*

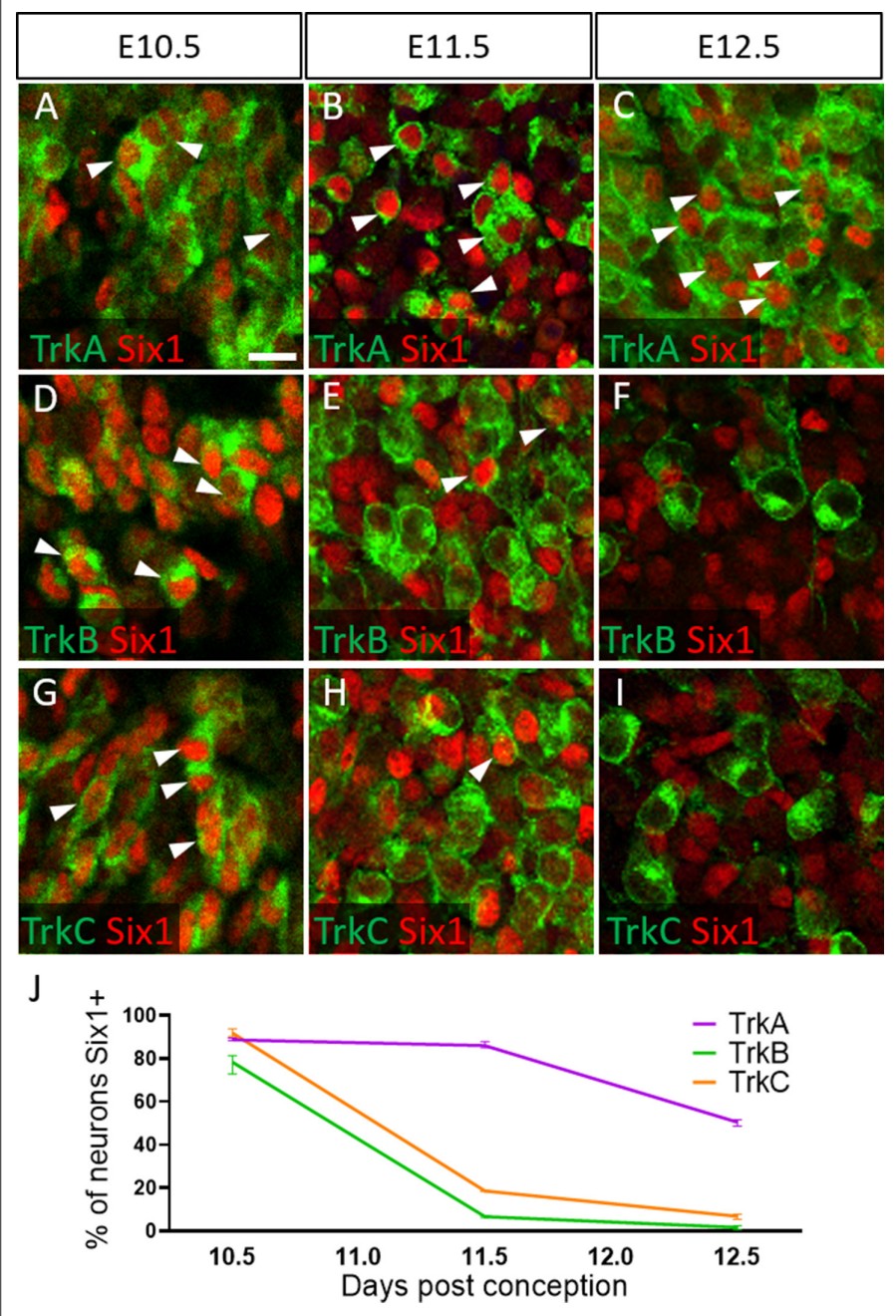

**Figure 6.** Dynamic expression of Six1 and Trk receptors occurs during trigeminal ganglion neurogenesis. (**A–I**) Fluorescent immunohistochemistry on representative horizontal sections at embryonic day 10.5 (E10.5) (A, D and G), E11.5 (B, E and H), and E12.5 (C, F and I) in Control embryos demonstrating expression of TrkA (A–C, green), TrkB (D–F, green), TrkC (G–I, green), and Six1 (A–I, red). Arrowheads point to neurons that co-express Six1 with TrkA (A–C), TrkB (D–F), or TrkC (G–I). (**J**) Quantification of the percentage of neurons expressing TrkA (purple), TrkB (green), or TrkC (orange) that also co-express Six1 in the Control trigeminal ganglion at E10.5 (n=2), E11.5 (n=3), and E12.5 (n=3). Data points represent mean ± SEM. Scale bars: 20 μm (A), applies to (B–I). Refer to *Figure 6— source data 1* for quantitative summary data represented in graphs.

The online version of this article includes the following source data and figure supplement(s) for figure 6:

**Source data 1.** Dynamic expression of Six1 and Trk receptors occurs during trigeminal ganglion neurogenesis.

**Figure supplement 1.** Normal developmental Trk expression dynamics in the trigeminal ganglion and maxillary nerve.

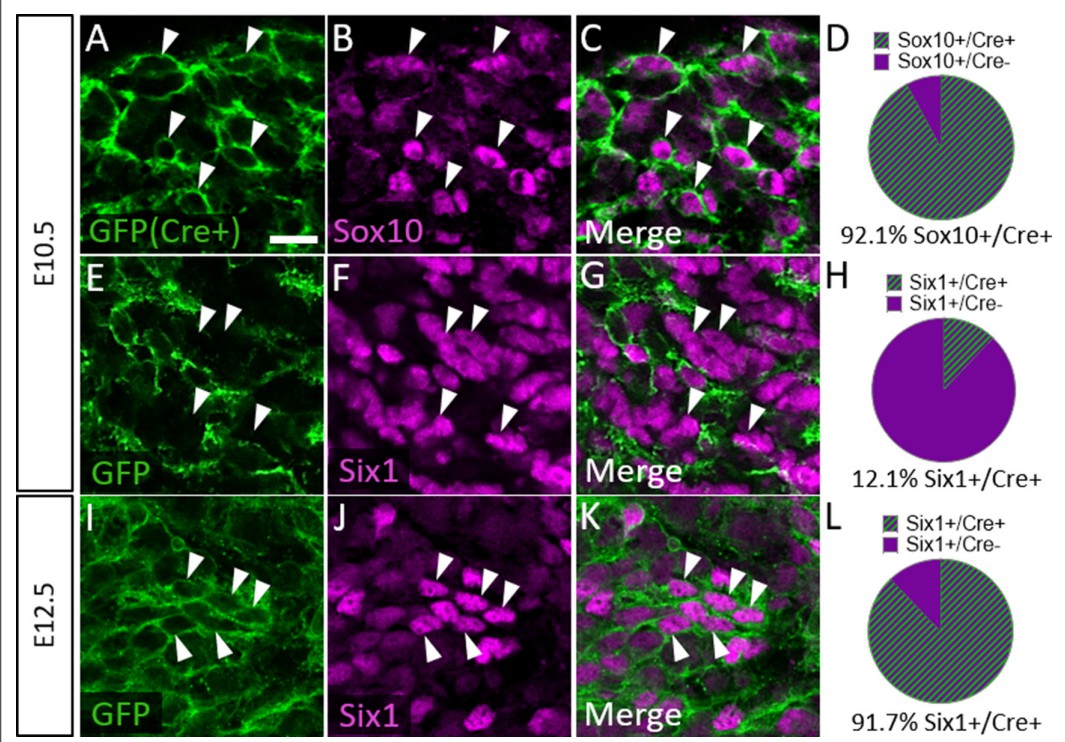

**Figure 7.** Six1 is first expressed by placodal neurons, followed by neural crest-derived neurons in the trigeminal ganglion. (**A–C, E–G, and I–K**) Representative images of fluorescent immunohistochemistry showing Sox10 (B and C, purple) or Six1 (F, G, J, and K, purple) with native green fluorescent protein (GFP) fluorescence in horizontal sections at embryonic day 10.5 (E10.5) (A–C and E–G, n=3) and E12.5 (I–K, n=3) in *Wnt1-Cre; ROSA^{mT/mG}* reporter embryos. Arrowheads point to neurons that co-express Sox10 (A–C) or Six1 (E–G and I–K) with GFP. (**D, H** and **L**) Pie charts demonstrating the percent of Sox10-positive (D) or Six1-positive (H and L) cells that co-express GFP in *Wnt1-Cre; ROSA^{mT/mG}* trigeminal ganglia at E10.5 (D and H) and E12.5 (L). Scale bars: 20 µm (A), applies to all images. Refer to *Figure 7—source data 1* for quantitative summary data represented in graphs.

The online version of this article includes the following source data for figure 7:

**Source data 1.** Six1 is first expressed by placodal neurons, followed by neural crest-derived neurons in the trigeminal ganglion.

*ROSA^{mT/mG}*) that expresses membrane-bound green fluorescent protein (GFP) in Wnt1-Cre-recombined cells and the red fluorescent protein TdTomato (hereafter referred to as RFP) in non-recombined cells (*Muzumdar et al., 2007*). In the *Wnt1-Cre; ROSA^{mT/mG}* mouse, 92.1% of Sox10-positive neural crest cells in the trigeminal ganglion expressed GFP at E10.5, indicating a large proportion of neural crest cells are targeted for Wnt1-Cre-mediated recombination (*Figure 7A–D*). Indeed, at E10.5, only 12.1% of Six1-expressing cells in the trigeminal ganglion were GFP-positive, confirming that the majority of Six1-positive cells are placode-derived at this stage (*Figure 7E–H*). In contrast, at E12.5, 91.7% of Six1-positive cells expressed GFP, indicating a shift to mostly neural crest-derived Six1-expressing cells at later stages of trigeminal ganglion neurogenesis (*Figure 7I–L*). Altogether, these results demonstrate that Six1 is a transient marker of newly differentiated neurons within the trigeminal ganglion and that the majority of Six1-expressing cells are neural crest-derived and express TrkA after E10.5.

We next examined Trk expression in the *Wnt1-Cre; ROSA^{mT/mG}* maxillary lobe, after the completion of trigeminal ganglion neurogenesis (E15.5). At this stage, the dense packing of neural crest-derived neurons and neural crest-derived satellite glia that express membrane-targeted GFP made it difficult to confidently quantify GFP-positive neurons. However, the non-recombined RFP-expressing population was less abundant and more easily discerned (*Figure 8A–D*). Since the vast majority of Sox10-expressing neural crest cells are recombined by Wnt1-Cre (*Figure 7A–D*), the RFP-expressing population within the trigeminal ganglion is, in theory, mostly placode-derived. At E15.5, only 24.4% of TrkA neurons, versus 77.4% of TrkB neurons and 78.6% of TrkC neurons, expressed RFP (*Figure 8E–P*).

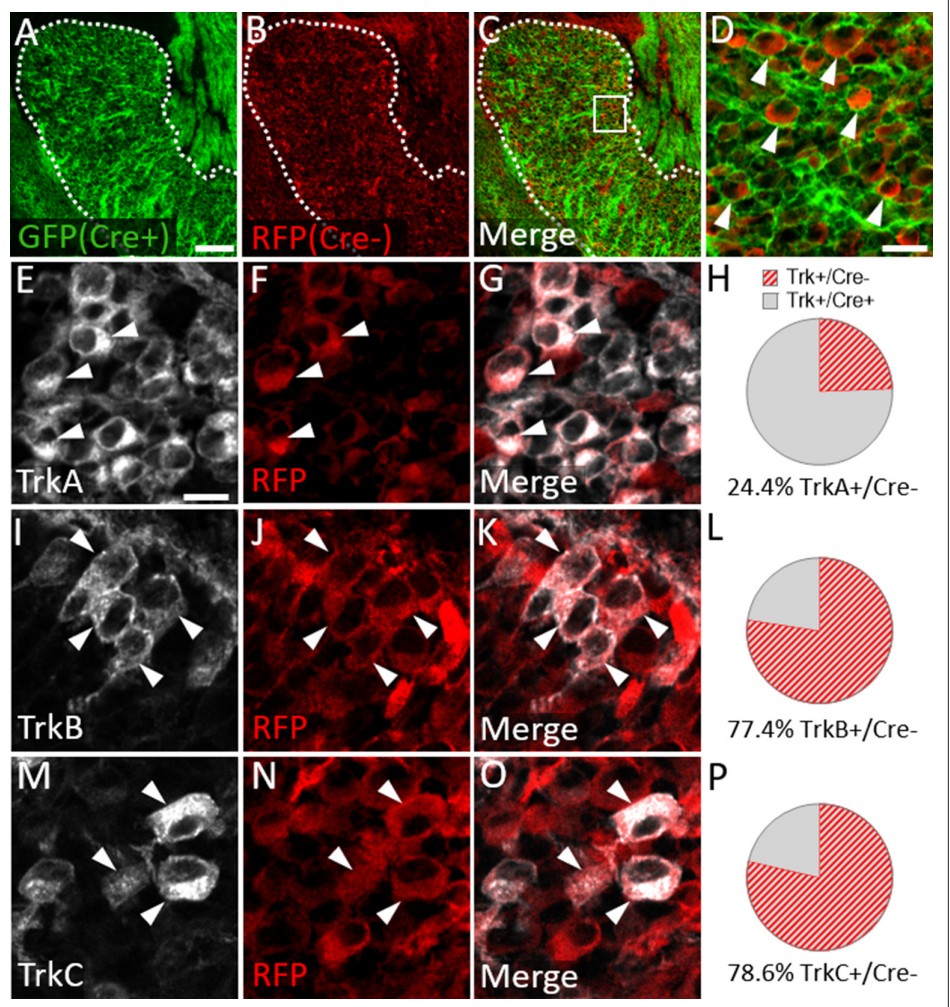

**Figure 8.** Neural crest-derived trigeminal ganglion neurons are biased to a TrkA fate, while placodal neurons express TrkB or TrkC. (**A–D**) Representative images of fluorescent immunohistochemistry at embryonic day (E15.5) showing red fluorescent protein (RFP; B–D) with native green fluorescent protein (GFP) fluorescence indicating Wnt1-Cre-mediated recombination (A, C and D) in horizontal sections through the maxillary lobe of the trigeminal ganglion in *Wnt1-Cre; ROSA^{mT/mG}* reporters. (D) Higher magnification of box in C. Arrowheads point to RFP-positive, non-recombined neurons. (**E–G, I–K, and M–O**) Fluorescent immunohistochemistry on serial sections through the maxillary lobe of *Wnt1-Cre; ROSA^{mT-mG}* reporters (n=3) showing TrkA (E and G, white), TrkB (I and K, white), or TrkC (M and O, white) with RFP (F, G, J, K, N, and O, red). Arrowheads point to neurons that express RFP and TrkA (E–G), TrkB (I–K), or TrkC (M–O). (**H, L and P**) Quantification of the percentage of neurons expressing TrkA (H), TrkB (L), or TrkC (P) that also co-express RFP at E15.5 (n=3). Scale bars: 100 μm (A), applies to (B–C); 20 μm (D); 10 μm (E), applies to (F, G, I–K, and M–O). Refer to *Figure 8—source data 1* for quantitative summary data represented in graphs.

The online version of this article includes the following source data for figure 8:

**Source data 1.** Neural crest-derived trigeminal ganglion neurons are biased to a TrkA fate, while placodal neurons express TrkB or TrkC.

These findings suggest that approximately three quarters of TrkA neurons in the embryonic trigeminal ganglion are neural crest-derived, while just one quarter of TrkB/C neurons are neural crest-derived. Thus, there is a strong correlation between cellular origin and eventual Trk receptor expression in the developing trigeminal ganglion, such that targeting neural crest cells will be more likely to affect TrkA versus TrkB or TrkC neurons, as seen in *Elp1* CKO.

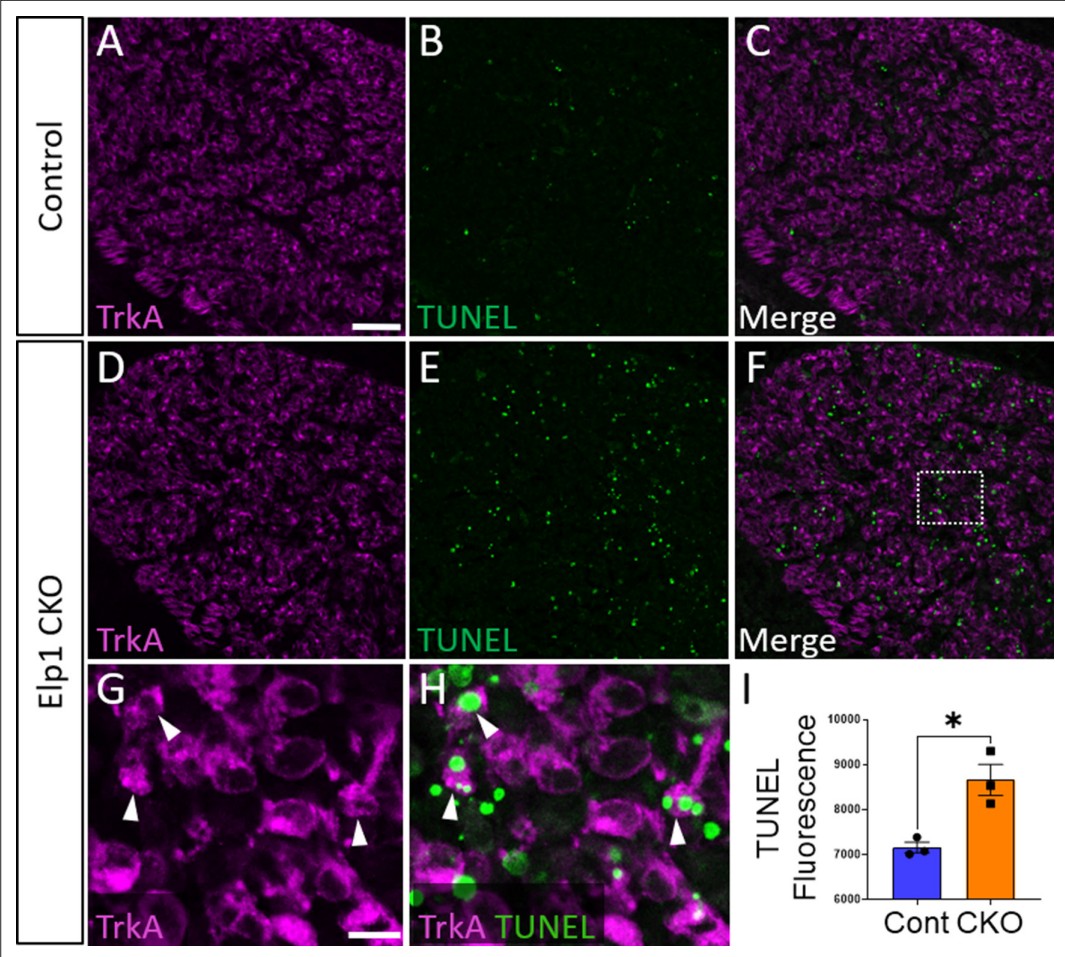

**Figure 9.** Aberrant apoptosis contributes to loss of neural crest-derived TrkA neurons in *Elp1* CKO trigeminal ganglia. (**A–H**) Fluorescent immunohistochemistry on representative horizontal sections from embryonic day 12.5 (E12.5) Control (A–C) or *Elp1* CKO (D–H) littermates revealing expression of TrkA (A, C, D, and F–H, purple) with TUNEL staining (B, C, E, F, and H, green). (G and H) Higher magnification of box in F. Arrowheads point to TrkA neurons that are TUNEL-positive (G and H). (**I**) Quantification of TUNEL fluorescence in Control (blue, 675 a.u., n=3) and *Elp1* CKO (orange, 8248 a.u., n=3, p=0.0052, nested unpaired t-test) trigeminal ganglia at E12.5. Values are mean ± SEM. *p=0.0224, unpaired t-test. a.u.: arbitrary units. Scale bars: 50 µm (A), applies to (B–F); 10 µm (G), applies to (H). Refer to *Figure 9—source data 1* for quantitative summary data represented in graphs.

The online version of this article includes the following source data and figure supplement(s) for figure 9:

**Source data 1.** Aberrant apoptosis contributes to loss of neural crest-derived TrkA neurons in Elp1 CKO trigeminal ganglia.

**Figure supplement 1.** TUNEL staining in glial progenitors and geniculate ganglia at embryonic day 12.5 (E12.5).

**Figure supplement 2.** Apoptosis in *Elp1* CKO is not a result of altered nerve growth factor (NGF) expression in target tissues.

## Loss of neural crest-derived TrkA neurons in *Elp1* CKO trigeminal ganglia results from aberrant apoptosis

To evaluate whether the loss of TrkA neurons is caused by aberrant cell death in the *Elp1* CKO trigeminal ganglion, TUNEL assays were performed with co-labeling for various markers. At E12.5, TUNEL fluorescent puncta were distributed throughout *Elp1* CKO trigeminal ganglia, and total TUNEL fluorescence was statistically higher in the E12.5 *Elp1* CKO trigeminal ganglion compared to Control, which only exhibited occasional, scattered TUNEL staining at this stage (*Figure 9*). Indeed, TUNEL-positive nuclei were often found in TrkA-expressing cells (*Figure 9G and H*). Notably, TUNEL fluorescent puncta were also observed in non-neuronal cells, so we assessed whether neural crest-derived

glial cells, too, were dying in the *Elp1* CKO trigeminal ganglion. Instead, we found that extra-neuronal TUNEL particles were found in the BFABP-positive cytoplasm of immature satellite glia and not in their Sox10-positive nuclei (*Figure 9—figure supplement 1A-D*). This suggests that glial precursors in the trigeminal ganglion phagocytose the cellular debris of nearby apoptotic sensory neurons, as previously reported in the trunk (*Wu et al., 2009*). Importantly, apoptosis of TrkA neurons was not explained by changes in the TrkA ligand, NGF, as NGF protein distribution in trigeminal nerve target tissues was similar between Control and *Elp1* CKO (*Figure 9—figure supplement 2*). Moreover, aberrant TUNEL staining was not observed in *Elp1* CKO geniculate ganglia compared to Control at E12.5, suggesting neural crest-derived neurons are most affected by apoptosis in *Elp1* CKO cranial ganglia at the examined timepoints (*Figure 9—figure supplement 1E-H*). Collectively, these findings suggest that deletion of *Elp1* from the neural crest lineage leads to innervation defects in the head and the selective loss of TrkA neurons as a result of widespread cell death within the trigeminal ganglion.

## Discussion

Since the discovery that *ELP1* mutations cause FD, diverse roles for Elp1 have been revealed in neuronal development, function, and degeneration (*Slaugenhaupt et al., 2001*; *Anderson et al., 2001*; *Dietrich and Dragatsis, 2016*; *Lefcort et al., 2017*). Many questions remain regarding the cellular and molecular mechanisms underlying FD phenotypes, particularly since Elp1 has been proposed to mediate several cytoplasmic (e.g. protein trafficking, α-tubulin acetylation, stress signaling, exocytosis, and tRNA modification and translation of codon-biased transcripts) and nuclear (i.e. transcription) functions (*Dalwadi and Yip, 2018*; *Dietrich and Dragatsis, 2016*; *Lefcort et al., 2017*). To date, animal studies emphasize defects in dorsal root, sympathetic, and enteric ganglia, which relay sensory and autonomic information from the trunk, limbs, and viscera to the CNS (*Abashidze et al., 2014*; *Cheng et al., 2015*; *George et al., 2013*; *Goffena et al., 2018*; *Hunnicutt et al., 2012*; *Jackson et al., 2014*; *Li et al., 2020*; *Morini et al., 2021*; *Tolman et al., 2022*). However, there is a dearth of knowledge about Elp1 in the cranial ganglia, even when clinical deficits strongly implicate cranial sensory dysfunction in FD (*Mendoza-Santiesteban et al., 2017*; *Barlow, 2009*; *Geltzer et al., 1964*; *Gutiérrez et al., 2015*; *Palma et al., 2018*; *Won et al., 2019*). Perhaps this is due to the complexity of cranial ganglion development; dorsal root, sympathetic, and enteric ganglia neurons are strictly neural crest-derived, whereas cranial ganglia contain neurons derived from both neural crest cells and ectodermal placodes. The trigeminal ganglion is unique, containing interspersed placode- and neural crest-derived neurons, whereas other cranial ganglia are comprised either with spatially segregated placode-derived neurons or neural crest-derived neurons, not both (*Blentic et al., 2011*; *D'Amico-Martel, 1982*; *D'Amico-Martel and Noden, 1983*; *Hamburger, 1961*; *Saint-Jeannet and Moody, 2014*; *Steventon et al., 2014*). In this study, we analyzed the morphological and cellular consequences of *Elp1* deletion from the neural crest lineage during trigeminal ganglion development, providing additional context to previously proposed mechanisms for Elp1 in trunk neural crest-derived neurons. Additionally, our findings reveal substantive insights regarding contributions of neural crest- versus placode-derived cells to specific neuron subpopulations within the trigeminal ganglion. Collectively, these results fill a critical knowledge gap in our understanding of trigeminal ganglion neurodevelopment and, notably, highlight that cranial nerve impairments in FD may arise from context-dependent defects in neural crest- and/or placode-derived neurons.

### Elp1 is enriched in trigeminal neurons during the course of neurogenesis and early innervation

We first examined Elp1 expression in the developing trigeminal ganglion using an *Elp1^LacZ* reporter mouse, in addition to immunohistochemistry on Control tissue (*Figure 1* and *Figure 1—figure supplement 1*). While diffuse β-galactosidase staining was observed throughout the trigeminal ganglion, Elp1 protein expression was more discrete, with robust expression in the cytoplasm of differentiated neurons (*Figure 1* and *Figure 1—figure supplement 1*). These results mirror previously reported patterns of Elp1 protein expression in differentiated trunk sensory and sympathetic neurons (*Hunnicutt et al., 2012*; *Jackson et al., 2014*; *Abashidze et al., 2014*). Little to no Elp1 protein was detected in developing satellite glial cells within the trigeminal ganglion or in Schwann cell precursors that line the embryonic nerves between E10.5 and E12.5, which encompass the normal period of trigeminal

ganglion neurogenesis and early innervation (*Figure 1* and *Figure 1—figure supplement 1*; *Huang et al., 1999b*; *Davies and Lumsden, 1984*; *Wilkinson et al., 1996*). While the anatomical distribution of Elp1 protein overlapped with β-galactosidase staining in *Elp1^LacZ* mice, the observation that Elp1 protein is more selectively expressed in neurons indicates Elp1 expression is potentially controlled at the post-transcriptional or post-translational level, such that Elp1 protein may be upregulated upon neuronal differentiation. Alternatively, β-galactosidase may have a longer half-life than Elp1, or Elp1 protein may be present in non-neuronal cells at levels below the threshold of detection via immuno-fluorescence. The presence of highly enriched Elp1 protein in neuronal cytoplasm, including axonal compartments, suggests there are extranuclear functions for Elp1 in trigeminal ganglion neurons, as noted in studies in other neuronal systems that localize Elp1 to axonal transport and synaptic vesicles (*Abashidze et al., 2014*; *Naftelberg et al., 2016*; *Tourtellotte, 2016*; *Li et al., 2020*).

## Elp1 is required for trigeminal nerve outgrowth and innervation of target tissues

To further explore trigeminal nerve deficits associated with FD, we examined trigeminal ganglion development using an established mouse model in which *Elp1* is conditionally deleted from neural crest cells and their derivatives (*George et al., 2013*). We found the initial formation and gross morphology of the trigeminal ganglion to be similar between *Elp1* CKO and Control embryos, with condensed placodal neurons and infiltrating neural crest cells both present in *Elp1* CKO at E10.5 (*Figure 2*). Although migration was not directly examined here, our findings complement conclusions that Elp1 is not required for trunk neural crest cell migration or subsequent formation of dorsal root or sympathetic ganglia (*Abashidze et al., 2014*; *George et al., 2013*; *Hunnicutt et al., 2012*; *Jackson et al., 2014*). Interestingly, knockdown of another elongator complex member, Elp3, inhibits neural crest cell migration in *Xenopus*, suggesting elongator subunits may have distinct individual functions in the neural crest (*Yang et al., 2016*).

As development proceeds, progressive defects in trigeminal nerve outgrowth and innervation are apparent in *Elp1* CKO embryos. By collecting high-resolution images of the intact ganglion and nerves over several timepoints, we observed various axonal abnormalities that precede this outcome. In the ophthalmic, maxillary, and mandibular divisions of the trigeminal nerve at E11.5, axons generally extended in the direction of their respective targets, but many deviated from established fascicles along the way, resulting in disorganized nerves with a 'hairy' appearance from ectopic branching (*Figure 3*). By E12.5, at least some trigeminal axons reached their respective target tissues (*Figure 4*). In the ophthalmic division of *Elp1* CKO, the frontal nerve initially extended branches around the eye (albeit less elaborately than in Controls, *Figure 4*) but then retracted by E13, resulting in an even less complex pattern of innervation (*Figure 5*). Therefore, at least some frontal nerve axons in *Elp1* CKO are able to reach their target site but are not maintained. In contrast, the medial and lateral nasal nerves (also ophthalmic division) never formed in *Elp1* CKO (*Figures 4 and 5*), indicating nasal nerve axons failed to navigate to their target destinations. These data suggest Elp1 is required for axonal outgrowth and proper target innervation in trigeminal ganglion neurons. Even more striking is the occurrence of unique phenotypes in distinct trigeminal nerve branches upon loss of Elp1 from neural crest derivatives, indicating several context-dependent mechanisms likely contribute to *Elp1* CKO and FD phenotypes.

## Elp1 is required for survival of TrkA neurons in the trigeminal ganglion

Given the drastic trigeminal nerve innervation deficits observed in *Elp1* CKO mice, we assessed the different Trk receptor-expressing neuron subpopulations in *Elp1* CKO and Control trigeminal ganglia. In *Elp1* CKO, we noted a decrease in TrkA immunoreactivity in the distal branches of the ophthalmic, maxillary, and mandibular nerves, as well as in the central nerve root and ganglion itself compared to Control (*Figure 5* and *Figure 5—figure supplement 1*). Additionally, there were fewer TrkA neurons in *Elp1* CKO trigeminal ganglia (*Figure 5*). This result was particularly intriguing since FD patients have reduced facial pain and temperature sensation, and TrkA-expressing sensory neurons generally develop into small-diameter nociceptors that relay pain and temperature signals (*Mu et al., 1993*; *Genç et al., 2005*; *Reichardt, 2006*).

A loss of TrkA neurons could potentially be explained by changes in neurogenesis or survival of TrkA-positive neurons. Reduced TrkA numbers in *Elp1* CKO dorsal root ganglia have been partially attributed

to early cell cycle exit and death of Pax3-positive TrkA progenitors due to reduced DNA repair (*George et al., 2013*; *Goffena et al., 2018*). In contrast, we observed no difference in the number of newly differentiated (Six1-positive) neurons in E11.5 *Elp1* CKO trigeminal ganglia. Moreover, Pax3-positive cells in the trigeminal ganglion express little to no Elp1 during the period of neurogenesis (*Figure 1—figure supplement 1*) and are reportedly glial, not neuronal, progenitors (*Baker et al., 2002*). Therefore, our findings are most consistent with the idea that loss of Elp1 from neural crest cells has major effects after neuronal differentiation, but we have not definitively ruled out neurogenic deficits.

Collectively, our data suggest that Elp1 is required for trigeminal ganglion axons to grow and properly invade target tissues in order to prevent neuronal apoptosis. Our observations complement previous studies showing Elp1 is required for the survival of TrkA neurons in sympathetic and dorsal root ganglia (*George et al., 2013*; *Jackson et al., 2014*; *Naftelberg et al., 2016*; *Li et al., 2020*; *Ohlen et al., 2017*). However, our discovery of *Elp1* CKO phenotypes at relatively early timepoints suggests there are additional functions for Elp1 during initial cranial sensory ganglion development. In sympathetic neurons, Elp1 is reported to modulate TrkA/NGF retrograde signaling by regulating the phosphorylation of TrkA receptors in signaling endosomes (*Li et al., 2020*). These studies were largely performed in cultured sympathetic neurons, with Elp1 expression perturbed perinatally, therefore bypassing the period of normal axon pathfinding and early innervation that we have examined here. Thus, our analyses of intact and sectioned embryos provide important context for the role of Elp1 in the development of trigeminal TrkA neurons *in vivo*.

Our findings neither confirm nor negate the possibility that Elp1 modulates TrkA/NGF signaling in some trigeminal ganglion neurons. For those axons that successfully navigate to intended targets, deficient TrkA retrograde signaling could lead to subsequent cell death. Indeed, the retraction of axons in the frontal nerve likely indicates an inability to receive and/or appropriately respond to target-derived neurotrophic support, especially considering we and others observe no changes in the expression of NGF in *Elp1* CKO target tissues (*Figure 9—figure supplement 2*, *George et al., 2013*; *Morini et al., 2016*; *Naftelberg et al., 2016*). However, the loss of TrkA immunoreactivity in distal axons (*Figure 5*) potentially contradicts this mechanism since the phosphorylation state of TrkA receptors and/or retrograde transport requires adequate TrkA expression in axon terminals (*Li et al., 2020*; *Naftelberg et al., 2016*). Moreover, we find that some cranial nerve branches, namely the medial and lateral nasal nerves, never form in *Elp1* CKO embryos, pre-empting any potential deficits in retrograde signaling. Given our findings, it will be critical to perform detailed spatiotemporal studies to gain a more complete understanding of TrkA and Elp1 sub-cellular expression and trafficking dynamics in trigeminal neurons and the role of Elp1 in these processes.

## Context-dependent functions for Elp1 exist during trigeminal ganglion neurodevelopment

The range of *Elp1* CKO phenotypes in different trigeminal nerve branches strongly implies cell type-dependent functions for Elp1, underscoring the inherent differences between distinct neuronal populations. This conclusion is further supported by contradictory observations that neuronal Elp1 depletion can increase or decrease neurite branching in varying contexts (*Hunnicutt et al., 2012*; *Abashidze et al., 2014*; *Jackson et al., 2014*; *Ohlen et al., 2017*). While opposing results may be due to variations in developmental timing or species, they may also highlight genuine differences in Elp1 function, depending upon the identity and environment of individual neurons.

For example, Elp1 may be required for axonal outgrowth via alternative, NGF-independent mechanisms, including anterograde trafficking of receptors for critical growth factors or guidance molecules and/or regulation of local protein synthesis, all of which are required for growing axons (*Scott-Solomon and Kuruvilla, 2018*; *Ascano et al., 2009*; *Cioni et al., 2019*; *Korsak et al., 2016*; *Batista and Hengst, 2016*; *Kang and Schuman, 1996*). Recently, Elp1 has been shown to regulate neuronal gene expression in a dose-dependent manner in a humanized mouse model of FD (*Morini et al., 2021*). Importantly, these functions of Elp1 could be direct or indirect via its essential role as part of the elongator complex in modifying tRNAs during translation (*Cameron et al., 2021*; *Chen et al., 2009b*; *Chen et al., 2009a*; *Goffena et al., 2018*; *Huang et al., 2005*; *Huang et al., 2008*; *Karlsborn et al., 2014*; *Karlsborn et al., 2015*). Indeed, the dorsal root ganglia proteome in *Elp1* CKO exhibits substantial changes (compared to Control) across a wide range of cellular functions, including axon guidance and pathfinding (*Goffena et al., 2018*).

In support of a potential role for Elp1 in axon pathfinding, we noted some degree of axon wandering along the ophthalmic, maxillary, and mandibular nerves in *Elp1* CKO embryos (*Figures 3 and 4*). Perhaps in these affected neurons, Elp1 is required for axons to respond to long-distance guidance cues. Alternatively, Elp1 may be required for inter-axonal adhesion or to limit axonal branching in order to maintain appropriate trajectories of newly growing axons along a pre-determined tract. While trigeminal ganglion target tissues express other neurotrophins (e.g. BDNF, NT-3, *Arumäe et al., 1993*; *Buchman and Davies, 1993*; *Ernfors et al., 1992*; *O'Connor and Tessier-Lavigne, 1999*), the axon defasciculation phenotype observed at E11.5 (*Figure 3*) is likely not due to axons trying to access these neurotrophins. In support of this, we note no further increase in defasciculation at later stages (E12.5 and E13). Given the phenotypes we observe are specific to TrkA neurons, we surmise that the defasciculated neurons are expressing TrkA. Use of alternative neurotrophins like BDNF or NT3, however, would require these neurons to switch receptor expression, and we do not see changes in TrkB or TrkC expression in *Elp1* CKO embryos. Instead, we speculate the defasciculation is tied, at least in part, to defects in adhesion among axons, particularly since *Elp1* CKO dorsal root ganglia have dramatically reduced levels of Cadherin-7 (*Goffena et al., 2018*). In chick cranial motor neurons, Cadherin-7 enhances axonal outgrowth and restricts interstitial axon branching (*Barnes et al., 2010*). Moreover, Elp1 plays a role in adhesion in other cell types (*Johansen et al., 2008*; *Cohen-Kupiec et al., 2011*). Whether similar *Elp1* CKO proteome changes or adhesive functions for Elp1 are conserved in the mouse trigeminal ganglion remains to be explored. Notably, with our current approach, it is difficult to attribute axon wandering to a specific cellular mechanism, that is, increased branching versus reduced adhesion, etc., due to the inability to isolate individual axons. In the future, it would be informative to apply recently published methods to analyze single axon trajectories of trigeminal ganglion neurons in *Elp1* CKO (*Motahari et al., 2020*).

## Insights into neural crest versus placodal neurogenesis in the developing trigeminal ganglion

Over the course of this study, we made several observations regarding the temporal sequence of events during trigeminal ganglion neurodevelopment. Although the ganglion contains a mixture of neural crest- and placode-derived neurons, which both express Elp1 (*Figure 1*), only neural crest-derived cells were targeted for *Elp1* deletion in these experiments. Since the vast majority of trigeminal ganglion neurons are placode-derived at E10.5 (*Figure 7*, *Karpinski et al., 2016*), it is not surprising that trigeminal ganglion neuroanatomy and cellular composition were similar between *Elp1* CKO and Control embryos at this stage (*Figure 2*). In contrast, striking alterations in the *Elp1* CKO trigeminal ganglion and nerves were noted after E10.5, with specific effects on TrkA-expressing neurons. We suspected this effect on TrkA neurons in our neural crest-specific knockout may be explained by the lineage of TrkA versus TrkB and TrkC neurons, rather than a TrkA-specific Elp1 mechanism.

While previous studies had defined Trk receptor expression and proportions of placode- versus neural crest-derived neurons in the E10.5 mouse trigeminal ganglion (*Huang et al., 1999a*; *Huang et al., 1999b*; *Wilkinson et al., 1996*; *Karpinski et al., 2016*), the dynamics of neural crest versus placodal neurogenesis after E10.5 were unclear. In the chick trigeminal ganglion, neural crest-derived cells are reportedly found proximal to the neural tube, whereas placodal neurons are situated in distal regions (*Steventon et al., 2014*). This pattern does not translate to the mouse trigeminal ganglion, which has been previously described as a mosaic of cellular subtypes with no preferential aggregation of any particular lineage (*Karpinski et al., 2016*; *Karpinski et al., 2022*). Therefore, anatomical position is an unreliable tool for predicting placodal versus neural crest lineage in the mouse trigeminal ganglion. Interestingly, studies in the chick trigeminal ganglion attribute large cell body diameter to potential placode lineage, while small diameter cells have been linked to neural crest lineage (*d'Amico-Martel and Noden, 1980*; *D'Amico-Martel and Noden, 1983*). Coincidentally, murine TrkB and TrkC neurons tend to have larger cell bodies compared to small-diameter TrkA-expressing neurons (*Fariñas et al., 1998*; *Huang et al., 1999a*). While this relationship has not been explicitly investigated in the mouse embryonic trigeminal ganglion, our data provide indirect evidence to support this link between cellular origin and size.

Our findings confirm that TrkB and TrkC neurons express Six1 and are the most abundant neurons in the trigeminal ganglion at E10.5 (*Figure 6* and *Figure 6—figure supplement 1*). Subsequently, the ratio of TrkA neurons to TrkB and TrkC neurons drastically increases by E12.5 (*Figure 6—figure*

*supplement 1*, *Huang et al., 1999a*; *Huang et al., 1999b*). After E10.5, TrkA neurons, by far, are the most likely to express the transcription factor Six1 (*Figure 6*), and at least 90% of Six1-positive cells are neural crest-derived (Wnt1-Cre-recombined) at this stage (*Figure 7*). These results strongly suggest that: (1) TrkB and TrkC neurons are born first from placodal precursors, (2) the majority of TrkA neurons arise later from neural crest progenitors, and (3) Six1 is transiently expressed in all newly differentiated trigeminal ganglion neurons, as it is in the dorsal root ganglion (*Yajima et al., 2014*). Therefore, Six1 is not a reliable marker of placodal lineage in the trigeminal ganglion after neural crest neurogenesis commences.

To provide experimental evidence for these assertions, we used the *Wnt1-Cre; ROSA^{mT/mG}* reporter mouse in which the majority of neural crest derivatives express GFP. We quantified the number of RFP-expressing (non-Wnt1-Cre-recombined, presumably placode-derived) neurons in the trigeminal ganglion after the conclusion of neurogenesis. At E15.5, approximately one quarter of TrkA neurons expressed RFP, while three quarters of TrkB and TrkC neurons expressed RFP (*Figure 8*). Our results complement a recent study which found that approximately two-thirds of TrkA neurons are Wnt1-Cre-recombined in the trigeminal ganglion of 8-day-old mice (TrkB and TrkC Wnt1-Cre ratios were not reported, *Karpinski et al., 2022*). Thus, these data indicate the Trk-expressing neuronal composition within the trigeminal ganglion is established during mid- to late-gestation and remains fairly constant into early postnatal life.

Intriguingly, *Karpinski et al., 2022* also reported targeted effects on TrkA neurons in the 22q11DS deletion mouse, which are preceded by premature neurogenic divisions of neural crest cells and increased variation of neuronal gene expression at E9.5 and E10.5, respectively. Using Six1 as a placode lineage marker, the authors concluded that greater proportions of Six1-positive cells and increased co-localization of Six1 and Wnt1-Cre in the trigeminal ganglion indicated a shift toward placodal differentiation within the 22q11DS trigeminal ganglion (*Karpinski et al., 2022*). Adopting the assumption that Six1 is expressed in newly differentiated neurons regardless of lineage, their findings can be reinterpreted instead through the lens of premature neural crest neurogenesis, leading to more Six1-positive cells and increased Six1/Wnt1-Cre co-localization and therefore would complement our conclusions. Moreover, the inclusion of (prematurely differentiated) neural crest-derived neurons along with placode-derived neurons could be an additional or alternative explanation for the increased variation in neuronal gene expression in the 22q11DS trigeminal ganglion at E10.5. Therefore, our findings are broadly relevant for the understanding of normal trigeminal ganglion neurogenesis and suggest that neural crest disorders and/or animal models which genetically target neural crest cells are likely to induce biased effects on TrkA neurons within the trigeminal ganglion.

In light of our conclusions, it follows that there were no observed differences in placode-derived TrkB or TrkC neurons in *Elp1* CKO embryos at the examined stages. Moreover, we did not detect similar neurodevelopmental perturbations in the exclusively placode-derived geniculate ganglion, which contains few, if any, TrkA-expressing neurons at E12.5. Given that Elp1 is expressed in placode-derived neurons (*Figure 1* and *Figure 5—figure supplement 2*), and recent reports showing Elp1 is required for the survival of TrkB-positive neurons in epibranchial ganglia (*Tolman et al., 2022*), we suspect a placode-targeted deletion of Elp1 would lead to TrkB neuronal deficits in the trigeminal and geniculate ganglia. It is also possible that placode-derived TrkB or TrkC neurons will be indirectly affected at later timepoints in *Elp1* CKO, given the important reciprocal interactions between the two lineages during trigeminal ganglion development (*Shiau et al., 2008*; *Shiau and Bronner-Fraser, 2009*; *Wu and Taneyhill, 2019*; *Steventon et al., 2014*; *Wu et al., 2014*). It would be informative to selectively delete *Elp1* from placode-derived trigeminal ganglion neurons only, or in addition to neural crest cells, in order to better understand the effects of cell type-specific loss of Elp1 on trigeminal ganglion development. Unfortunately, there are no trigeminal placode-specific Cre drivers available at this time.

Remarkably, the pattern of nerve deficits in *Elp1* CKO embryos provides indirect evidence that certain branches of trigeminal sensory nerves may also be specifically neural crest- or placode-derived. For instance, innervation of the whisker pad is initiated by E11.5 in both Control and *Elp1* CKO (*Figure 3*). However, the innervation field does not increase substantially in the *Elp1* CKO (*Figures 4 and 5*). Surviving nerve terminals express very little TrkA, whereas Control embryos develop a more expansive, TrkA-abundant innervation pattern by E13 (*Figure 5*). This suggests that placode-derived neurons may reach the whisker pad first and act as scaffolds upon which later-differentiating TrkA

axons can traverse. This is supported by distinct Trk receptor expression in the maxillary nerve at E11, in which the vast majority of early maxillary nerve fibers express TrkB or TrkC, with few initial contributions of TrkA-positive axons (*Figure 6—figure supplement 1*). In contrast, some trigeminal nerve branches never form in the *Elp1* CKO, namely the medial and lateral nasal nerves of the ophthalmic division (*Figures 4 and 5*), begging the question of whether these nerves are exclusively composed of axons from neural crest-derived neurons. While we hoped to assess this in the *Wnt1-Cre; ROSA*^mT/mG reporter mouse, the presence of GFP-expressing Schwann cell precursors along the embryonic nerves unfortunately hindered analysis using this approach. Thus, more in-depth studies are required to understand the diversity of interactions between neural crest- and placode-derived cells during trigeminal ganglion neurogenesis and innervation and to understand the nuanced roles that Elp1 has in such a complex system.

An additional limitation to our studies is that the Wnt1-Cre driver does not target every neural crest cell. Our results are similar to previous reports in the trunk (*Hari et al., 2012*), in that upward of 90% of Sox10-positive neural crest cells undergo Wnt1-Cre-mediated recombination in the trigeminal ganglion (*Figure 7*). However, a population of non-recombined neural crest derivatives is certainly present in the trigeminal ganglion of *Elp1* CKO and *Wnt1-Cre; ROSA*^mT/mG reporter mice, which may potentially skew our analyses. Little is known about potential subpopulations of neural crest precursors that may exist within the trigeminal ganglion, whether they differentially contribute to neuronal or glial subpopulations, or whether they are preferentially recombined by Wnt1-Cre. Nonetheless, Wnt1-Cre-mediated recombination is commonly used to target neural crest cells (*Debbache et al., 2018*) and achieves more efficient recombination than other drivers, including Sox10-Cre (*Jacques-Fricke et al., 2012*). Since the precedent for previous FD animal studies is Wnt1-Cre-mediated deletion of *Elp1*, our current model is useful for the initial characterization of effects on the trigeminal ganglion. There are marked differences, though, in the *Elp1* CKO and the presentation of FD in humans. *Elp1* CKO phenotypes are more extreme due to the complete loss of Elp1 from neural crest derivatives, whereas some Elp1 protein is still present in neurons of FD patients (*Slaugenhaupt et al., 2001*). Moreover, it is currently unclear at the cellular level whether certain neuronal subpopulations are more greatly affected in FD patients. In the future, use of alternative Cre drivers or humanized FD mice may provide additional insight into yet undiscovered mechanisms in FD.

In summary, FD is characterized by a wide range of sensory and autonomic phenotypes, including those that implicate involvement of the cranial trigeminal ganglion and its associated nerves. Through studies in a conditional knockout mouse model that eliminate Elp1 from the neural crest lineage, we have uncovered distinct effects on the development of trigeminal sensory neurons, specifically those expressing TrkA that correlate with nociceptive function. Altogether, these findings highlight defects in cranial nerve development that may contribute to loss of facial pain and temperature sensation in FD.

## Materials and methods
### Animal husbandry

All animal care and use described herein were in accordance with federal and institutional guidelines and approved by Montana State University's and University of Maryland's IACUC, under protocols #2018–81 (MSU) and #R-MAR-20–15 (UMD). The generation of the *Elp1* conditional knockout mouse was described previously by Dr. Frances Lefcort (Montana State University) who generously provided embryos and mice for this work (*George et al., 2013*). All mice were created and maintained on a C57BL/6 background. *Elp1*^fl/fl mice have *LoxP* sites flanking the coding region of the fourth exon of *Elp1* (previously *Ikbkap*). When the floxed region is excised via Cre recombinase, the resulting truncated *Elp1* transcript is eliminated from cells by nonsense-mediated decay. The role of Elp1 in Wnt1-expressing neural crest cells and derivatives was examined by crossing homozygous *Elp1*^fl/fl mice with hemizygous *Wnt1-Cre*^+/- mice (The Jackson Laboratory, stock no. 003829) to create *Elp*^fl/+;*Wnt1-Cre*^+/- males, which were then crossed with *Elp1*^fl/fl females to generate *Elp1* conditional knockout mice (*Elp1*^fl/fl; *Wnt1-Cre*^+, abbreviated '*Elp1* CKO' throughout) and a 1:3 ratio. For all analyses, *Elp1* CKO mice were compared to littermate Controls (*Elp1*^fl/+; *Wnt1-Cre*^-), and at least two litters were examined per experiment. Genotyping was performed via PCR using the primer sequences listed below. *Elp1*^LacZ reporter mice, in which *LacZ* is targeted to the *Elp1* locus between the third and fourth exon

(*George et al., 2013*), were a gift from Dr. Frances Lefcort. *ROSA^{mT/mG}* reporter mice that express membrane-targeted GFP in Cre-recombined cells and TdTomato (a type of RFP) in non-recombined cells were purchased from the Jackson Laboratory (stock #007676) and crossed with *Wnt1-Cre^+* mice for neural crest lineage tracing experiments. Genotype was determined by the presence of GFP in the embryo when viewed under a fluorescent light source. For timed breeds, the day of the vaginal plug was considered E0.5. Samples sizes were determined based on previously published studies using this mouse model or similar *Elp1* conditional knockouts mouse models (*George et al., 2013*; *Jackson et al., 2014*; *Li et al., 2020*).

## Genotyping

Genomic DNA was extracted using the Extracta DNA Prep for PCR kit (Quantabio) according to manufacturer instructions. *Elp1* CKO and Control alleles were detected with the following primers: 5'-GCACCTTCACTCCTCAGCAT-3' (forward) and 5'-AGTAGGGCCAGGAGAGAACC-3' (reverse). The *Wnt1-Cre* allele was detected with the following primers: 5'-GCCAATCTATCTGTGACGGC-3' (forward) and 5'- CCTCTATCGAACAAGCATGCG-3' (reverse). PCR mixtures were prepared using DreamTaq Green PCR Master Mix (Thermo Fisher) according to the manufacturer's protocol.

## Tissue collection and preparation

Timed pregnant females were euthanized via $CO_2$ asphyxiation followed by cervical dislocation. Embryos were collected and placed in ice-cold 1× PBS. A hindlimb bud was collected from each embryo for genotyping. Embryos were fixed via submersion and gentle shaking in 4% paraformaldehyde/1× PBS for between 20 min (E10.5) and 90 min (E13.5) at room temperature, then rinsed three times in 1× PBS for 20 min each. Fixed embryos were stored in 1× PBS with 0.02% sodium azide at 4°C until further analysis. For sectioning, embryos were rinsed twice with 1× PBS, then submerged in 15% sucrose (w/v) in 1× PBS at 4°C overnight, or until tissue sank, followed by submersion in 30% sucrose at 4°C until tissue sank. Embryos were first equilibrated in a 1:1 solution of 30% sucrose/1× PBS and Tissue-Tek optimal cutting temperature compound (OCT, Fisher) for 2 hr at 4°C and then in 100% OCT at 4°C for 2 hr. Embryos were embedded in 100% OCT using liquid nitrogen vapor and stored at –80°C, followed by sectioning at 12 μm on a cryostat (Leica) and collection of tissue sections on Superfrost Plus charged slides (VWR).

## Immunohistochemistry

### Tissue sections

A hydrophobic boundary was drawn around tissue sections using an ImmEdge Pen (Vector Labs). Tissue sections were rehydrated with 1× PBS for 5 min, then permeabilized with 1× PBS/0.5% Triton X-100 (Tx-100) for 5 min at room temperature. Tissue was blocked with 5% bovine serum albumin (BSA, Fisher Scientific) (w/v) in PBS-Tx (1× PBS, 0.1% Tx-100) for approximately 1 hr at room temperature, then rinsed once in PBS-Tx. When using the mouse anti-RFP antibody, mouse Fab fragments were added to the blocking solution at a concentration of 1:40 (Jackson Immuno). Primary antibodies were diluted in PBS-Tx plus 1% BSA and applied overnight at 4°C in a humidified chamber. Unbound primary antibodies were washed off with four PBS-Tx rinses for 5 min each at room temperature. Sections were then incubated with secondary antibodies, diluted in PBS-Tx plus 1% BSA, for 1 hr at room temperature in a humidified chamber. Sections were rinsed three times in PBS-Tx for 5 min each, followed by two rinses in 1× PBS for 5 min each, all at room temperature. Coverslips were mounted with DAPI Fluoromount-G Mounting Medium (Southern Biotech) and allowed to dry in the dark at room temperature overnight before imaging.

### Whole-mount

Fixed embryos were rinsed twice with 1× PBS for 5 min per rinse, then dehydrated through a series of increasingly concentrated methanol (MeOH) washes (50:50, 80:20, 100:0 MeOH:PBS) for 30 min each at room temperature. Embryos were incubated in Dent's Bleach (4:1:1 MeOH:DMSO:30% $H_2O_2$) for 6 hr at room temperature with gentle shaking, then rehydrated through a series of decreasingly concentrated MeOH washes (50:50, 20:80, 0:100 MeOH:PBS) for 30 min each at room temperature. For blocking, embryos were incubated in antibody dilution solution (1× PBS, 0.1% Tx-100, and 5% BSA) for at least 2 hr at room temperature. Next, embryos were incubated with fresh antibody dilution

solution containing primary antibodies for 2 days (E10.5) up to 4 days (E13.5) at 4°C with gentle shaking. Embryos were washed four times for ~1 hr each at room temperature with PBS-Tx, then incubated in fresh dilution solution with secondary antibodies for 1 day (E10.5) up to 3 days (E13.5) at 4°C with gentle shaking. Embryos were washed three times for ~1 hr each at room temperature with PBS-Tx, followed by two washes with 1× PBS for 30 min each at room temperature. E10.5 embryos were imaged at this step, whereas older embryos were cleared before imaging, as described below.

## Antibodies

Primary antibodies used included the following: Elp1 (Sigma #SAB2701068, 1:500), Tubb3 (Abcam #ab78078, 1:1,000 for sections, 1:300 for whole-mount), Sox10 (R&D #AF2864, 1:200 or GeneTex #GTX128374, 1:500), TrkA (R&D #AF1056, 1:500 for sections, 1:200 for whole-mount), TrkB (R&D #AF1494, 1:300), TrkC (R&D #AF1404, 1:300), Six1 (Sigma #HPA001893, 1:500), Islet1 (DSHB #PCRP-ISL1-1A9, 1:500), Neuropilin2 (R&D cat. AF567, 1:500), Pax3 (DSHB, 'Pax3', 1:200), BFABP (Sigma, ZRB13190-25ul, 1:500), and RFP (Thermo, MA515257, 1:400). All species/isotype-specific, Alexa Fluor-conjugated secondary antibodies were purchased from Thermo Scientific and used at a dilution of 1:500 on sections or 1:300 in whole-mount.

## FRUIT clearing

After whole-mount immunohistochemistry, embryos were subjected to FRUIT clearing (*Hou et al., 2015*). Briefly, embryos were moved through a series of aqueous FRUIT buffers, containing 8 M urea (Sigma), 0.5% (v/v) α-thioglycerol (TCI America), and increasing concentrations of fructose (Fisher). Embryos were incubated at room temperature with gentle rocking in 35% FRUIT for 6 hr, followed by 40% FRUIT overnight, 60% FRUIT for 8 hr, and 80% FRUIT overnight. Embryos were stored in 80% FRUIT at 4°C until imaging in this buffer.

## TUNEL staining

TUNEL staining was performed on tissue sections after immunohistochemistry using the In Situ Cell Death Detection Kit, TMR Red (Roche) according to the manufacturer's instructions. After washing off unbound secondary antibodies, slides were post-fixed with 4% paraformaldehyde in 1× PBS for 5 min at room temperature, then washed twice with 1× PBS for 5 min each at room temperature. Sections were incubated with TUNEL reaction mixture for 60 min at 37°C in the dark, followed by three washes in 1× PBS for 5 min each at room temperature. Coverslips were mounted using DAPI Fluoromount-G Mounting Medium (Southern Biotech) and allowed to dry in the dark at room temperature overnight before imaging.

## Imaging

E10.5 embryos that underwent whole-mount β-galactosidase staining or fluorescent immunohistochemistry were imaged on a Zeiss SteREO Discovery V8 Pentafluor fluorescent microscope using AxioVision software (Zeiss). Embryos E11.5 and older that were processed for whole-mount immuno-histochemistry were imaged in 80% FRUIT buffer on a Zeiss LSM 800 confocal microscope. Z-stacks were collected at 5 μm intervals using 5× or 10× air objectives. Fluorescent immunohistochemistry on tissue sections was also visualized on the Zeiss confocal microscope using 10× and 20× air objectives, or the 63× oil objective. For all applications, laser power, gain, offset, and digital zoom were identical when imaging comparable regions of interest in Control versus *Elp1* CKO embryos and the pinhole was set to 1 airy scan unit at all times. CZI files were processed in Zen software and histograms adjusted identically for Control versus *Elp1* CKO tissue within experimental groups (Blue edition 2.0, Zeiss). For Z-stacks, CZI files were processed in ImageJ, where maximum intensity projections were created using the Z-Project function in Hyperstack mode.

## Nerve tracing and Sholl analysis

Tracing of the frontal nerve was performed on TIFF images of maximum intensity Z-stack projections, described above, using the Simple Neurite Tracer plug-in in Image J (NIH). After tracing, the Sholl analysis function within the Simple Neurite Tracer plug-in was used to quantify nerve branching complexity under the following settings: use standard axes, no normalization of intersections, and 10 μm radius step size. The center point was set on the primary frontal nerve branch, just below the

first branch point. Individual distributions and group average distributions were plotted together in Microsoft Excel (n=2 Control, n=4 *Elp1* CKO, from two different litters).

## Cell number, ganglion size, central root, and nerve quantification

For cell number quantification, 20× images were acquired from horizontal sections in the maxillary lobe of the trigeminal ganglion. The boundary of the trigeminal ganglion was determined in each section by Tubb3 or Sox10 staining, then a region of interest (ROI) was created using FIJI and applied to the respective images for analysis. Using the Cell Counter plug-in in FIJI, each cell of interest was counted within the ROI. Multiple sections were quantified per animal. Statistical comparisons were performed in Graphpad Prism as 'nested analyses'. Control and *Elp1* CKO groups were compared by unpaired nested t-tests and adjusted for multiple comparisons, where appropriate, using the Holm-Sidak method. Other anatomical measurements were performed on Tubb3-labeled maximum intensity Z-projections in FIJI using the Line and Measure functions. To measure the size of the trigeminal ganglion, a straight line was drawn in FIJI from the top to bottom of the ganglion perpendicular to the maxillary nerve equidistant from the neural tube. Similarly, to measure the extent of the infraorbital nerve in the whisker pad, a straight line was drawn perpendicular to the maxillary nerve, spanning the greatest length from top to bottom of Tubb3-labeled nerve endings. To measure nerve length, a line was drawn along the nerves with the Line Segment tool from the ganglion exit point to the visible end of the nerve. Groups were compared by unpaired t-tests in Graphpad Prism and adjusted for multiple comparisons using the Holm-Sidak method.

## Fluorescence measurements

The 20× unsaturated fluorescent images were acquired from horizontal sections in the maxillary lobe of the trigeminal ganglion. The boundary of the trigeminal ganglion was determined by Tubb3 or Sox10 staining, then a ROI was created using FIJI (*Schindelin et al., 2012*) and applied to the respective images for analysis. Using the Measure function in FIJI, the mean pixel gray value was quantified within each ROI. Multiple sections were quantified per animal. Statistical comparisons were performed in Graphpad Prism as 'nested analyses'. Control and *Elp1* CKO groups were compared by unpaired nested t-tests.

## Key resources table

| Reagent type (species) or resource | Designation | Source or reference | Identifiers | Additional information |
|---|---|---|---|---|
| Genetic reagent (*Mus musculus*) | *Elp1*<sup>LacZ</sup> (formerly referred to as "*Ikbkap:LacZ*") | *George et al., 2013* | | Gifted by Frances Lefcort |
| Genetic reagent (*Mus musculus*) | *Wnt-1/GAL4/cre-11* | *Danielian et al., 1998* | RRID:IMSR_JAX:003829 | Gifted by Frances Lefcort |
| Genetic reagent (*Mus musculus*) | *Elp1*<sup>fl/fl</sup> | *George et al., 2013* | | Gifted by Frances Lefcort |
| Genetic reagent (*Mus musculus*) | *ROSA*<sup>mT/mG</sup> | The Jackson Laboratory | RRID:IMSR_JAX:007676 | |
| Antibody | anti-Elp1 (Rabbit polyclonal) | Sigma | Cat# SAB2701068 | IF (1:500) |
| Antibody | anti-β-tubulin III (Mouse monoclonal) | Abcam | Cat# ab78078 | IF (1:1,000 sections, 1:300 whole-mount) |
| Antibody | anti-Sox10 (Goat polyclonal) | R&D | Cat#: AF2864 | IF (1:200) |
| Antibody | anti-Sox10 (Rabbit polyclonal) | GeneTex | Cat# GTX128374 | IF (1:500) |
| Antibody | anti-TrkA (Goat polyclonal) | R&D | Cat# AF1056 | IF (1:500 sections, 1:200 whole-mount) |
| Antibody | anti-TrkB (Goat polyclonal) | R&D | Cat#: AF1494 | IF (1:300) |
| Antibody | anti-TrkC (Goat polyclonal) | R&D | Cat#: AF1404 | IF (1:300) |
| Antibody | anti-Six1 (Rabbit polyclonal) | Sigma | Cat# HPA001893 | IF (1:500) |

*Continued on next page*

*Continued*

| Reagent type (species) or resource | Designation | Source or reference | Identifiers | Additional information |
|---|---|---|---|---|
| Antibody | anti-Islet1 (Mouse monoclonal) | DSHB | Cat#PCRP-ISL1-1A9 | IF (1:500) |
| Antibody | anti-Neuropilin2 (Goat polyclonal) | R&D | Cat#: AF567 | IF (1:500) |
| Antibody | anti-Pax3 (Mouse monoclonal) | DSHB | Cat#: "Pax3" | IF (1:200) |
| Antibody | anti-BFABP (Rabbit monoclonal) | Sigma | Cat#: ZRB13190-25ul | IF (1:500) |
| Antibody | anti-RFP (Mouse monoclonal) | Thermo Fisher | Cat#: MA515257 | IF (1:400) |
| Antibody | anti-NGF (Rabbit polyclonal) | Abcam | Cat#: ab52918 | IF (1:200) |
| Sequence-based reagent | Elp1 floxed allele (forward) | *George et al., 2013* | PCR primers | GCACCTTCACTCCTCAGCAT |
| Sequence-based reagent | Elp1 floxed allele (reverse) | *George et al., 2013* | PCR primers | AGTAGGGCCAGGAGAGAACC |
| Sequence-based reagent | Wnt1-Cre allele (forward) | *George et al., 2013* | PCR primers | GCCAATCTATCTGTGACGGC |
| Sequence-based reagent | Wnt1-Cre allele (reverse) | *George et al., 2013* | PCR primers | CCTCTATCGAACAAGCATGCG |
| Commercial assay or kit | TMR Red TUNEL Kit | Sigma | Cat#: 12156792910 | To detect apoptotic cells |
| Software and algorithm | FIJI | Open Source | RRID:SCR_002285 | For image analyses |
| Software and algorithm | Zen Blue | Zeiss | RRID:SCR_013672 | For image analyses |
| Other | Donkey anti-mouse Fab fragments | Jackson Immuno | Cat#: 715-007-003 | IF (1:40 in blocking solution); see "Immunohistochemistry", "Tissue Sections" |
| Other | Fluoromount-G with DAPI | Southern BioTech | Cat#: 0100–20 | Mounting media for slides; see "Immunohistochemistry", "Tissue Sections" |

# Acknowledgements

We thank Marta Chaverra for assistance with collecting embryos, Lynn George for guidance regarding genetics, Vickie Riojas for help with animal husbandry, and Jenn Lafrican for technical assistance.

# Additional information

## Competing interests

Frances Lefcort: is the Co-Chair of the Scientific Advisory Board of the Familial Dysautonomia Foundation, Inc. The other authors declare that no competing interests exist.

## Funding

| Funder | Grant reference number | Author |
|---|---|---|
| National Institutes of Health | | Lisa A Taneyhill Frances Lefcort |

The funders had no role in study design, data collection and interpretation, or the decision to submit the work for publication.

## Author contributions

Carrie E Leonard, Conceptualization, Data curation, Formal analysis, Investigation, Methodology, Writing – original draft, Writing – review and editing; Jolie Quiros, Formal analysis, Investigation,

Validation; Frances Lefcort, Methodology, Resources, Writing – review and editing; Lisa A Taneyhill, Conceptualization, Funding acquisition, Methodology, Project administration, Resources, Supervision, Writing – original draft, Writing – review and editing

## Author ORCIDs

Carrie E Leonard  http://orcid.org/0000-0002-2199-8637
Frances Lefcort  http://orcid.org/0000-0002-2064-8678
Lisa A Taneyhill  http://orcid.org/0000-0002-8630-2514

## Ethics

All animal care and use described herein was in accordance with federal and institutional guidelines and approved by Montana State University's and University of Maryland's IACUC, under protocols #2018-81 (MSU) and #R-MAR-20-15 (UMD).

## Decision letter and Author response

Decision letter https://doi.org/10.7554/eLife.71455.sa1
Author response https://doi.org/10.7554/eLife.71455.sa2

---

## Additional files

### Supplementary files

• Transparent reporting form

### Data availability

All data generated or analysed during this study are included in the manuscript and supporting files. Source Data files have been provided for Figures 2-9.

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
