## [Editor Report]

This study uses a combination of conditional knockout mouse embryos with targeted deletion of Elp1 in neural crest cells and neuron-specific antibodies to identify the onset of neural defects associated with the trigeminal ganglion. This manuscript will be of interest to developmental biologists studying neurodevelopment disorders and provides important insights into the mechanisms underlying Familial Dysautonomia in the cranial sensory ganglia.

---

## [Decision Letter]

**Decision letter after peer review:**

Thank you for submitting your article "Loss of Elp1 disrupts trigeminal ganglion neurodevelopment in a model of Familial Dysautonomia" for consideration by *eLife*. Your article has been reviewed by 3 peer reviewers, and the evaluation has been overseen by Marianne Bronner as the Senior Editor. The reviewers have opted to remain anonymous.

Essential revisions:

The reviewers agree that this is an interesting topic and that the paper has potential. However, they ask for better quantification of the phenotypes and more functional data. In addition, the identity of a Vg neuron as derived from neural crest versus placode needs further study as no marker is perfect. Comparison to a placode-only derived ganglion (e.g. VIIIg) would add a level of experimental comparison that would be informative and novel compared to the previous investigations of DRGs. I refer you to the full reviews below for further details.

*Reviewer #1 (Recommendations for the authors):*

• p8, para1, line7: please make the term "the eye" more specific – does this include the lens, retina, cornea?

• Figure 1 legend: line 8: Carets for Isl1 indicate positive neuronal nuclei – the way the sentence is worded one could read is as cell body. Same for Figure 1 supplemental legend – Isl1 in neuronal nuclei not cell bodies.

• Figure 2: these are very nice schematics; how were they constructed?

• Figure 3: perhaps label the NT and VIIg in A and B.

• p18, para1, line8: is the protein more "selectively expressed" or "most abundantly expressed"? It may be that Elp1 turns over faster in some cells compared to the long-lived LacZ or it is expressed in the other cells but not at levels easily detected by IF.

• I am not crazy about using the term "model" in the title of the paper because Elp1 is only knocked down in the neural crest, which is not the case in the patients. Or do the aberrant nerves reported herein model the nerve deficiencies seen in patients?

*Reviewer #2 (Recommendations for the authors):*

1. In characterizing Elp1 expression during gangliogenesis, the authors use several markers to identify the placode and neural crest-derived neurons in regions where they appear to overlap. However, these neurons are segregated in the proximal (neural crest) and distal (placode) regions of the ganglion, which in addition to the markers they used would make a stronger point. The authors could also use similar location of neurons in addition to differential expression of TrKA/TrKB to confirm the absence of Elp1 in the neural crest-derived neuron as opposed to the placode-derived neurons of the CKO mice, and also to show that neuron apoptosis occurs in only the distal region of the trigeminal ganglion. Furthermore, the authors could use this differential expression of TrKA and TrKB to show the specific loss of TrKA neurons in the target sites of the Elp1 CKO mice.

2. The authors state that two litters were examined per experiment, but do not provide the numbers of knockout and wild type mice used for each experiment. In addition, quantification of the data such as the thickness of the central nerve root between control and Elp1 CKO mice would make the authors claim stronger.

3. The authors conclude that nerve defects caused by loss of Elp1 could be based on the target region. What is the expression pattern of NGF in the target tissues where the nerve bundles appear to be disformed in Elp1 CKO? Does the NGF expression overlap with other neurotrophic factors in this region?

4. The authors suggest that in Elp1 CKO mice, most TrKA neurons do not express Elp1. Since this was knocked out in NC-derived neurons, these results would be much stronger of they showed that Elp1 expression is maintained in the placode (TrkB) neurons under these conditions.

5. Are there specific targets innervated by neural crest- or placode-derived neurons? Given the defects observed in Elp1 CKO, what happens to the placode-derived neurons (TrKB neurons) that should be functioning normally?

Other concerns

1) Abstract: last sentence; The authors state that "These findings explain the loss of facial pain and temperature sensation in FD." Since this study did not evaluate facial pain, I suggest that they rephrase this sentence to something like this "These findings explain the defects in cranial gangliogenesis that may lead to loss of facial pain and temperature sensation in FD."

2) Introduction: Page 5; The authors introduce the small-diameter TrKA neurons but not the large-diameter TrKB neurons. Making this distinction in the introduction section would be helpful for the readers that are new to this topic.

3) Results section: Page 8; Based on the β-galactosidase expression, the authors conclude that Elp1 is expressed in cranial neural tissues. Given that there is vivid expression in what appears to be the neuroretina, I suggest that they report this instead of grossly stating that it is expressed in the eye.

4) Results section: Page 9; (Figure 1K-M) does not show Sox10-positive glia. Also Figure 1N-P are not addressed in the Results section.

5) Results section: Page 12, last sentence; The authors cannot claim to observe significant loss of TrKA without any quantification.

6) Results section: Page 13; The authors state that "…while the remaining Elp1-positive neurons had large cell bodies characteristic of TrkB/C neurons". It is not clear from these results whether these are the large cell bodied neurons. Co-staining with TrkB/C would clarify this statement.

7) Discussion section: Page 18; Based on their observation that the ganglion size appears not to be affected during early gangliogenesis, the authors infer that migration of cranial neural crest is not affected. For clarity, they should not state that Elp1 is not necessary for NC migration without direct results showing neural crest migration.

8) Discussion section: Page 19; Could the authors discuss the possibility that the defasciculation observed in the Elp1 CKO be due to axons trying to access neurotrophic factors from other sources other than the NGF that is potentially expressed in the target tissues?

9) Discussion section: Page 22; The statement "…deficient TrkA retrograde signaling could result in dying axons and subsequent cell death." is confusing. Please clarify.

10) Figure 2. This figure can be placed in the Supplementary section and/or broken up in the subsequent figures with each schematic depicting the developmental stage being analyzed.

11) Figure 6. Abbreviations for Na (OpV) and WP are missing.

*Reviewer #3 (Recommendations for the authors):*

The paper by Leonard et al. is a nice, concise analysis of the wnt-cre elp1 mouse with respect to trigeminal nerve development. The manuscript is well written and organized, the scientific questions are clearly stated, and the data answer the questions. The weaknesses of this paper lay in the scope and brad design.

– The story is very descriptive

– There is somewhat of a lack of mechanistic insight beyond the description of the development in this mouse. The only connection to the human disorder is based on described symptoms of facial pain and temperature insensitivity. There are reports in human systems, for example based on iPSC cell work that show different results (for example neural crest migration). The fact that this paper only investigates the WNT-CKO animal may make the authors miss alternative interpretations that may be important for the human disorder.

– The data very heavily relies on staining and morphology. Other techniques could strengthen the results, i.e. PCR for gene expression.

– The authors quote 'we deduced that TrkA neurons in the trigeminal ganglion must be primarily neural crest-derived, while TrkB and TrkC neurons are placode-derived – a concept that has been alluded to in the literature, but not explicitly demonstrated'. This is an interesting novel finding, can this be strengthened by additional assessment models or techniques?

---

## [Author Response]

Essential revisions:The reviewers agree that this is an interesting topic and that the paper has potential. However, they ask for better quantification of the phenotypes and more functional data. In addition, the identity of a Vg neuron as derived from neural crest versus placode needs further study as no marker is perfect. Comparison to a placode-only derived ganglion (e.g. VIIIg) would add a level of experimental comparison that would be informative and novel compared to the previous investigations of DRGs. I refer you to the full reviews below for further details.Reviewer #1 (Recommendations for the authors):• p8, para1, line7: please make the term "the eye" more specific – does this include the lens, retina, cornea?• Figure 1 legend: line 8: Carets for Isl1 indicate positive neuronal nuclei – the way the sentence is worded one could read is as cell body. Same for Figure 1 supplemental legend – Isl1 in neuronal nuclei not cell bodies.• Figure 2: these are very nice schematics; how were they constructed?• Figure 3: perhaps label the NT and VIIg in A and B.• p18, para1, line8: is the protein more "selectively expressed" or "most abundantly expressed"? It may be that Elp1 turns over faster in some cells compared to the long-lived LacZ or it is expressed in the other cells but not at levels easily detected by IF.• I am not crazy about using the term "model" in the title of the paper because Elp1 is only knocked down in the neural crest, which is not the case in the patients. Or do the aberrant nerves reported herein model the nerve deficiencies seen in patients?

We thank the Reviewer for these suggestions, which have been incorporated into the revised manuscript. The schematics were hand drawn by the first author using an iPad and the Procreate app. We concede that there are aspects of this conditional knockout animal that do not align with the human disorder of Familial Dysautonomia (FD) and have now expanded on these caveats in the Discussion. Nonetheless, this conditional knockout and others with similar limitations have previously been described by leading FD researchers as “animal models of FD”. As *eLife* specifically urges authors to name the model animal used in the title, we have not modified the title at this time but will defer to the Editor on her preference.

Reviewer #2 (Recommendations for the authors):1. In characterizing Elp1 expression during gangliogenesis, the authors use several markers to identify the placode and neural crest-derived neurons in regions where they appear to overlap. However, these neurons are segregated in the proximal (neural crest) and distal (placode) regions of the ganglion, which in addition to the markers they used would make a stronger point. The authors could also use similar location of neurons in addition to differential expression of TrKA/TrKB to confirm the absence of Elp1 in the neural crest-derived neuron as opposed to the placode-derived neurons of the CKO mice, and also to show that neuron apoptosis occurs in only the distal region of the trigeminal ganglion. Furthermore, the authors could use this differential expression of TrKA and TrKB to show the specific loss of TrKA neurons in the target sites of the Elp1 CKO mice.

We appreciate the point the Reviewer brings up regarding cell position within the trigeminal ganglion, particularly given our own research on chick trigeminal ganglion assembly. In the chick trigeminal ganglion, neurons are segregated by cellular origin such that placode-derived neurons reside in the distal ganglion (relative to the neural tube), while neural crest-derived neurons reside in the proximal ganglion. This pattern does not translate to the mouse trigeminal ganglion, which has been previously described as a mosaic of cellular subtypes with no preferential aggregation of any particular lineage (Karpinski et al., 2016; Motahari et al., 2021). Therefore, anatomical position is an unreliable tool for predicting placodal versus neural crest lineage in the mouse trigeminal ganglion. Our TrkA/B/C immunohistochemistry data, and the distribution of TrkA/TUNEL-double positive cells within Elp1 CKO trigeminal ganglion sections, also support this finding. Because of this, we cannot rely on position as another means to support our results. We have now addressed these differences between chick and mouse in the Discussion.

We have conducted additional section immunohistochemistry experiments in which we have co-stained for different Trks and Elp1 in control and Elp1 CKO embryos (Figure 5, Figure 5-supplement 1 and 2, E12.5). We discovered a statistically significant decrease in TrkA fluorescence intensity in Elp1 CKO versus control trigeminal ganglia, with no change observed for TrkB or TrkC. Additionally, there were fewer TrkA-expressing nerve endings in sections through the upper lip of Elp1 CKO embryos relative to control embryos, with no change observed in TrkC-expressing nerve endings. Remarkably in the Elp1 CKO, most TrkA neurons were devoid of Elp1 protein, while the majority of Elp1-positive neurons expressed TrkB or TrkC. Altogether, these data provide further evidence to support targeting of presumptive neural crest-derived TrkA neurons in the trigeminal ganglion upon Elp1 loss in neural crest cells.

2. The authors state that two litters were examined per experiment, but do not provide the numbers of knockout and wild type mice used for each experiment. In addition, quantification of the data such as the thickness of the central nerve root between control and Elp1 CKO mice would make the authors claim stronger.

We apologize for the omission of these numbers and have added them to the manuscript. In addition, we have now quantified several aspects of trigeminal ganglion and nerve development (described in detail in Reviewer 1, Point #5 above), as per the recommendations of this Reviewer and Reviewer 1. We thank the Reviewer for this comment, as the new data have strengthened the manuscript. Additionally, the numbers of animals/litters per experiment have been clarified in figure legends, and graphs representing statistical comparisons between control and Elp1 CKO embryos now include individual data points against the mean/SEM to demonstrate the number of embryos examined and the variation within genotypes.

3. The authors conclude that nerve defects caused by loss of Elp1 could be based on the target region. What is the expression pattern of NGF in the target tissues where the nerve bundles appear to be disformed in Elp1 CKO? Does the NGF expression overlap with other neurotrophic factors in this region?

The Reviewer brings up an excellent point. To address this, we have performed immunohistochemistry to examine NGF expression and distribution in Elp1 CKO and control littermates at E12.5 (Figure 9-supplement 2). Our data reveal NGF protein in trigeminal nerve target tissues of both control and Elp1 CKO embryos. Importantly, given the role of Elp1 in translation, these findings demonstrate that Elp1 is not altering NGF protein levels. These results are consistent with previously published data showing no difference in the amount of *NGF* transcripts (Naftelberg et al., 2016; Morini et al., 2021) or protein (George et al., 2013) levels between control and Elp1 CKO. With regards to other neurotrophic factors, BDNF and NT-3, which can serve as ligands for TrkB and TrkC, respectively, are also expressed in the ophthalmic and maxillary regions targeted by the trigeminal nerves (Ernfors et al., 1992; Arumäe et al., 1993; Buchman et al., 1993; O’Conner and Tessier-Lavigne, 1999).

4. The authors suggest that in Elp1 CKO mice, most TrKA neurons do not express Elp1. Since this was knocked out in NC-derived neurons, these results would be much stronger of they showed that Elp1 expression is maintained in the placode (TrkB) neurons under these conditions.

We thank the Reviewer for this suggestion. We now provide data from the Elp1 CKO showing that TrkA neurons are typically devoid of Elp1 protein, while TrkB and TrkC neurons still express Elp1 (Figure 5-supplement 2; see also Point #1 in this section).

5. Are there specific targets innervated by neural crest- or placode-derived neurons? Given the defects observed in Elp1 CKO, what happens to the placode-derived neurons (TrKB neurons) that should be functioning normally?

The Reviewer raises an excellent question regarding target tissues innervated by neural crest- vs. placode-derived neurons emanating from branches of the trigeminal ganglion. We are actively pursuing this line of research in my lab but it is currently beyond the scope of this study given the mouse work and time required to rigorously interrogate this question. We have speculated about this in the Discussion as a future direction. As a foray into this, however, we have quantified Trk fluorescence at E12.5 and find no statistically significant difference in TrkB (or TrkC) fluorescence throughout the trigeminal ganglion between control and Elp1 CKO embryos (Figure 5), while TrkA fluorescence is reduced. Additionally, we show that TrkA nerve endings are reduced in sections through the whisker pad of E12.5 Elp1 CKO embryos, but TrkC nerve endings are maintained (Figure 5).

6) Abstract: last sentence; The authors state that "These findings explain the loss of facial pain and temperature sensation in FD." Since this study did not evaluate facial pain, I suggest that they rephrase this sentence to something like this "These findings explain the defects in cranial gangliogenesis that may lead to loss of facial pain and temperature sensation in FD."

We thank the Reviewer for this suggestion and have modified the sentence accordingly.

7) Introduction: Page 5; The authors introduce the small-diameter TrKA neurons but not the large-diameter TrKB neurons. Making this distinction in the introduction section would be helpful for the readers that are new to this topic.

We appreciate this comment by the Reviewer and have revised the Introduction to include this information.

8) Results section: Page 8; Based on the β-galactosidase expression, the authors conclude that Elp1 is expressed in cranial neural tissues. Given that there is vivid expression in what appears to be the neuroretina, I suggest that they report this instead of grossly stating that it is expressed in the eye.

We thank the Reviewer for this suggestion and have altered the Results section.

9) Results section: Page 9; (Figure 1K-M) does not show Sox10-positive glia. Also Figure 1N-P are not addressed in the Results section.

We apologize for this mistake. We have revised the text to address these omissions.

10) Results section: Page 12, last sentence; The authors cannot claim to observe significant loss of TrKA without any quantification.

We thank the Reviewer for pointing this out. We have now quantified TrkA, TrkB, and TrkC immunofluorescence within the trigeminal ganglion in E12.5 tissue sections and note a statistically significant decrease in TrkA fluorescence, but no change in TrkB or TrkC fluorescence, in Elp1 CKO versus control (Figure 5).

11) Results section: Page 13; The authors state that "…while the remaining Elp1-positive neurons had large cell bodies characteristic of TrkB/C neurons". It is not clear from these results whether these are the large cell bodied neurons. Co-staining with TrkB/C would clarify this statement.

This is an excellent suggestion by the Reviewer. To provide further clarity, we have now co-stained Elp1 CKO sections through the trigeminal ganglion for TrkB (or TrkC) and Elp1, revealing that while most TrkA neurons are devoid of Elp1, the majority of the remaining neurons express TrkB and TrkC (Figure 5-supplement 2).

12) Discussion section: Page 18; Based on their observation that the ganglion size appears not to be affected during early gangliogenesis, the authors infer that migration of cranial neural crest is not affected. For clarity, they should not state that Elp1 is not necessary for NC migration without direct results showing neural crest migration.

We apologize for this overinterpretation of our data, which was also pointed out by Reviewer 1 (Point #3). Thus, we have removed this language from the manuscript.

13) Discussion section: Page 19; Could the authors discuss the possibility that the defasciculation observed in the Elp1 CKO be due to axons trying to access neurotrophic factors from other sources other than the NGF that is potentially expressed in the target tissues?

We appreciate this insight from the Reviewer. As mentioned in this section in Point #3, other neurotrophins are expressed in trigeminal ganglion target tissues, including BDNF and NT-3, which can serve as ligands for TrkB and TrkC, respectively. However, we think it is unlikely that the defasciculation phenotype we observe at E115 is due to axons trying to access these other neurotrophins. In support of this, we note no further increase in defasciculation at later stages (E12.5, E13) in the Elp1 CKO maxillary branch, providing indirect evidence that the axons are not actively seeking different neurotrophic factors for their support. Given the phenotypes we observe are specific to TrkA neurons, we surmise that the defasciculated neurons are expressing TrkA. Use of alternative neurotrophins like BDNF or NT3, however, would require these neurons to switch receptor expression, and we do not see any changes in TrkB or TrkC expression in Elp1 CKO embryos over developmental time. Moreover, the absence of any increase in defasciculation at later stages (by putative TrkA neurons) can be correlated with the increased apoptosis of TrkA neurons in Elp1 CKO embryos. Instead, we speculate that the defasciculation is tied, at least in part, to defects in adhesion among axons, particularly since Elp1 CKO dorsal root ganglia have dramatically reduced levels of Cadherin-7 (Goffena et al., 2018). In chick cranial motor neurons, Cadherin-7 enhances axonal outgrowth and restricts interstitial axon branching (Barnes et al., 2010). Whether similar *Elp1 CKO* proteome changes or Cadherin-7 functions are conserved in the mouse trigeminal ganglion remains to be explored. Moreover, Elp1 plays a role in adhesion in other cell types (Johansen et al., 2008; Cohen-Kupiec et al., 2010). We have now added this text to the Discussion and thank the Reviewer for raising this interesting point.

14) Discussion section: Page 22; The statement "…deficient TrkA retrograde signaling could result in dying axons and subsequent cell death." is confusing. Please clarify.

We apologize for the confusion and have revised the text to instead read “…deficient TrkA retrograde signaling could lead to subsequent cell death.”

15) Figure 2. This figure can be placed in the Supplementary section and/or broken up in the subsequent figures with each schematic depicting the developmental stage being analyzed.

We thank the Reviewer for this suggestion. We have now broken up the cartoon images in Figure 2 and added them to their relevant figures.

16) Figure 6. Abbreviations for Na (OpV) and WP are missing.

We apologize for this oversight. We have added these labels to the cartoon and images in Figure 4, which has now replaced Figure 6, using “Io” (infraorbital nerve) instead of “WP” (whisker pad).

Reviewer #3 (Recommendations for the authors):The paper by Leonard et al. is a nice, concise analysis of the wnt-cre elp1 mouse with respect to trigeminal nerve development. The manuscript is well written and organized, the scientific questions are clearly stated, and the data answer the questions. The weaknesses of this paper lay in the scope and brad design.– The story is very descriptive.– The data very heavily relies on staining and morphology. Other techniques could strengthen the results, i.e. PCR for gene expression.

The Reviewer brings up an important point that was raised by the other Reviewers of the manuscript. To further bolster our findings, we have now quantified various aspects of the forming trigeminal ganglion and its nerve branches in Elp1 CKO and control embryos (see Reviewer 1, Point #5). Given the TrkA-specific deficits in the trigeminal ganglion, we also evaluated NGF, the ligand for TrkA, and found no change in its distribution in Elp1 CKO compared to control. Finally, we generated a Wnt1-Cre;ROSA^mT/mG^ reporter mouse in which all neural crest derivatives (Wnt1-Cre-recombined) express GFP, while other cell types like placode cells express RFP. The use of this new mouse model allowed us to answer questions related to the cellular origin of different Trk-expressing neuronal subpopulations in the trigeminal ganglion and to define Six1 as a marker for all newly differentiating neurons in the trigeminal ganglion, irrespective of cellular origin. We hope the Reviewer finds these additional experiments to strengthen the manuscript and deem it suitable for publication.

– There is somewhat of a lack of mechanistic insight beyond the description of the development in this mouse. The only connection to the human disorder is based on described symptoms of facial pain and temperature insensitivity. There are reports in human systems, for example based on iPSC cell work that show different results (for example neural crest migration). The fact that this paper only investigates the WNT-CKO animal may make the authors miss alternative interpretations that may be important for the human disorder.

We thank the Reviewer for his/her suggestion and have addressed this caveat in the Discussion. However, others have shown that the Sox10-Cre driver does not target all cranial neural crest cells (see Debbache et al., 2018; Jacques-Fricke et al., 2012; Hari et al., 2012). Therefore, we may miss certain populations of neural crest cells if we switch to this line. Importantly, Wnt1-Cre-mediated deletion of *Elp1* is the precedent for previous FD animal studies, making our model appropriate for initial characterization of effects on the trigeminal ganglion. The use of alternative Cre drivers in future studies may provide unique insights into currently unknown mechanisms underlying FD, which is beyond the scope of our study.

– The authors quote 'we deduced that TrkA neurons in the trigeminal ganglion must be primarily neural crest-derived, while TrkB and TrkC neurons are placode-derived – a concept that has been alluded to in the literature, but not explicitly demonstrated'. This is an interesting novel finding, can this be strengthened by additional assessment models or techniques?

This is an excellent suggestion by the Reviewer that was also put forth by Reviewer 1 (Point #7). We have addressed this by crossing our Wnt1-Cre line with the ROSA^mT/mG^ reporter. As detailed in Point #1 above, we used this reporter mouse to distinguish neural crest-derived neurons and glia (green) from placode-derived neurons, which remain red. We performed section immunohistochemistry using antibodies to the different Trks and find that most TrkB- and C-expressing neurons are placode-derived, while neural crest cells give rise to the majority of TrkA-expressing neurons (Figure 8).